# Estimating Radar Precipitation in Cold Climates: The role of Air Temperature within a Nonparametric Framework

Kuganesan Sivasubramaniam[1], Ashish Sharma[2], and Knut Alfredsen[1]

[1]Department of Civil and Environmental Engineering, Norwegian University of Science and Technology, 7491 Trondheim, Norway
[2]School of Civil and Environmental Engineering, University of New South Wales, Sydney, NSW2052, Australia

*Correspondence to:* Kuganesan Sivasubramaniam (kuganesan.sivasubramaniam@ntnu.no)

**Abstract.** The use of ground based precipitation measurements in radar precipitation estimation is well known in radar hydrology. However, the approach of using gauged precipitation and near surface air temperature observations to improve radar precipitation estimates in cold climates is much less common. In cold climates, precipitation is in the form of snow or rain or a mixture of the two phases. Air temperature is intrinsic to the phase of the precipitation and could therefore be a possible covariate in the models used to ascertain radar precipitation estimates. In the present study, we investigate the use of air temperature within a nonparametric predictive framework to improve radar precipitation estimation for cold climates. A nonparametric predictive model is constructed with radar precipitation rate and air temperature as predictor variables, and gauge precipitation as an observed response using a k-nearest neighbour (k-nn) regression estimator. The relative importance of the two predictors is ascertained using an information theory-based weighting. Four years (2011-2015) of hourly radar precipitation rate from the Norwegian national radar network over the Oslo region, hourly gauged precipitation from 68 gauges, and gridded observational air temperature were used to formulate the predictive model and hence make our investigation possible. Gauged precipitation data were corrected for wind induced undercatch before using them as true observed response. The predictive model with air temperature as an added covariate reduces root mean squared error (RMSE) by up to 15 % compared to the model that uses radar precipitation rate as the sole predictor. More than 80 % of gauge locations in the study area showed improvement with the new method. Further, the associated impact of air temperature became insignificant at more than 85 % of gauge locations when the near surface air temperature was warmer than 10° C, which indicates that the partial dependence of precipitation on air temperature is most useful for colder temperatures.

## 1 Introduction

Hydrological applications require accurate precipitation estimates at the catchment scale (Beven, 2012; Kirchner, 2009). Weather radars provide quantitative precipitation estimates over a large area with high spatial and temporal resolution. However, weather radars measure the precipitation rate indirectly, using the energy scattered back by hydrometeors in the volume illuminated by a transmitted electromagnetic beam (Villarini and Krajewski, 2010a). The backscattered energy is measured as reflectivity which is used to estimate precipitation (Hong and Gourley, 2015).

The nature of radar precipitation measurements is subject to many sources of error. Some of the known errors in the reflectivity measurement are ground clutter, beam blocking, anomalous propagation, bright band, hail, and attenuation (Berne and Krajewski, 2013; Chumchean et al., 2003). During the conversion, the use of inappropriate Z - R relationship leads to Z - R conversion error. Due to the presence of such significant errors (both random and systematic), radar data are still not widely used in hydrological applications (Berne and Krajewski, 2013; Chumchean et al., 2003). Many studies (e.g., Abdella, 2016; Villarini et al., 2008; Ciach et al., 2007; Chumchean et al., 2006) have focused on estimating these errors in order to improve quantitative radar precipitation estimates; however, some of the underlying physical processes are still not understood well enough to allow significant advances (Villarini and Krajewski, 2010b).

In the standard approach, radar measurements of reflectivity ($Z$) are converted into precipitation rate ($R$) using the parametric Z - R relationship derived by Marshall and Palmer (1948) in the form of a power law, $Z = aR^b$. The variability of the power law parameters (a and b) is related to a number of factors including the drop size distribution (DSD) of hydrometeors. Drop size distribution varies in time and space as well as for the type and the phase of precipitation (Chumchean et al., 2008; Joss et al., 1990; Uijlenhoet, 2001; Wilson and Brandes, 1979).

In cold climates, precipitation occurs in the form of snow or rain or a mixture of snow and rain. Several studies (e.g., Battan, 1973; Sekhon and Srivastava, 1970; Marshall and Gunn, 1952) have investigated the Z - R relationship regarding the precipitation phase and proposed different parameter sets. Most radar operations in cold climate countries (e.g., Canada and Finland) use two sets of Z - R relations, one for rain and one for snow, often calibrated in situ to measure a water equivalent radar reflectivity factor (Koistinen et al., 2004; Crozier et al., 1991; Smith, 1984). However, the Norwegian radars and European radar project OPERA have used a single Z - R relationship (Marshall and Palmer (1948) relation for rain ($Z = 200R^{1.6}$)) throughout the year. The use of the single reflectivity-precipitation relationship can result in phase dependent bias in radar precipitation estimation.

The Finnish Meteorological Institute devised two equations for rain ($Z = 316R^{1.5}$) and snow ($Z_e = 100S^2$) for operational use (Saltikoff et al., 2015). Here $Z_e$ represents the equivalent radar reflectivity factor of snow. For the use of phase dependent reflectivity-precipitation (Z - R) relationship, the precipitation phase of the radar pixel must be estimated. Air temperature has traditionally been used to determine the phase of the precipitation (Al-Sakka et al., 2013). The Finnish Meteorological Institute uses temperature and humidity observations from synoptic stations to estimate the precipitation phase and uses that information to apply a different parameter set for rain or snow (Koistinen et al., 2004; Saltikoff et al., 2015). However, Saltikoff et al. (2000) reported that real time phase dependent adjustment of two different parameter sets does not improve the snowfall estimates significantly. To account for varying precipitation phase (multiple snow types and mixture of snow and rain), many parameter sets could be required. Moreover, the precipitation phase changes rapidly even within the single winter storm and hence, operationally, switching between different parameter sets can be a challenging task (Koistinen et al., 2004; Saltikoff et al., 2015).

Fassnacht et al. (2001, 1999) demonstrate the use of surface air temperature to estimate the fraction of snow content in mixed precipitation and use it to adjust the radar estimates for mixed precipitation. It was reported that this adjustment improved the

accumulated snow estimates in Ontario, Canada. Further Fassnacht et al. (1999) showed that the adjusted radar data provided more realistic precipitation estimates for precipitation-runoff models than corrected gauge precipitation data.

Starting from its origin and throughout its entire journey, the rain drop or snow crystal is shaped by temperature. During the formation and growth of cloud droplets, different temperatures and the degree of super saturation cause different shapes of crystals to form, and then the crystals start to fall. The falling crystals are then characterised by the temperature of the air through which they fall. As a result, the air temperature determines the final properties and the phase of the hydrometeor that reaches the ground surface (Fassnacht et al., 2001). Further, studies showed that there are multiple snow types with different shapes and densities and they vary in time, based partially on temperature (Saltikoff et al., 2015). Many studies (Auer Jr, 1974; Kienzle, 2008; Killingtveit, 1976; Rohrer, 1989) examined the relationship between the precipitation phases (snow, rain and mixture of snow and rain) and temperature. The probability of occurrence of snowfall versus temperature shows generally an approximately 'S' shaped structure and in some cases linear relation (Fassnacht et al., 2013) in these studies. Further, the dielectric property of solid particles (ice) is not the same as liquid particles (water) and moreover, it varies with temperature (Joss et al., 1990). These imply that temperature is intrinsic to both the phase of precipitation and the ensuing conversion of reflectivity into the incident ground precipitation.

Parametric (or regression type) and nonparametric approaches (nearest neighbour and kernel density estimation) have been used to build predictive models for a range of applications. A key advantage of nonparametric approaches is that less rigid assumptions about the distribution of the observed data are needed (Silverman, 1986) and hence no major assumptions about the process being modelled are required to construct the complete predictive system (Mehrotra and Sharma, 2006; Sharma and Mehrotra, 2014).

Ciach et al. (2007) used a nonparametric kernel regression to model radar rainfall uncertainty. They described the relation between true rainfall and radar-rainfall as the product of a systematic distortion function along with a random component and presented procedures to identify the two components. The distortion function could account for systematic biases which can be mathematically defined as a conditional expectation function, while the random component accounts for random errors in radar rainfall estimation. Villarini et al. (2008) estimated the conditional expectation function (distortion function) using both nonparametric (similar to Ciach et al. (2007)) and copula-based methods and compared the difference in performance between the two approaches using different quality metrics. It was found that performance of the nonparametric method was comparable with the copula-regression estimate and even outperformed when Nash Sutcliffe Efficiency (NSE) was used as a quality metric. The strength of nonparametric approaches is the ability to adapt to the data locally and the weakness is that the method results in "local" biases as a result of outliers (Villarini et al., 2008). Hasan et al. (2016a) presented a kernel based nonparametric approach to estimate ground rainfall using radar reflectivity as a univariate predictor variable in a tropical setting. Past observed radar reflectivity and gauged rainfall were used in formulating the nonparametric model.

In this study, the hypothesis is that near surface air temperature observations can help improve radar precipitation estimates in cold climates. Here, the nonparametric approach of Hasan et al. (2016a) can be extended to allow use of a bivariate predictor vector with air temperature as an additional predictor variable to precipitation. This forms the basis for the investigation reported in this study.

This study set out to investigate the use of air temperature as an additional predictor in the radar precipitation estimation with the objective of improving quantitative radar precipitation estimation for cold climates. Compared to traditional radar-gauge adjustment, the proposed method is based on nonparametric approach using gauge precipitation and air temperature observations to adjust the radar precipitation. The precipitation estimates using a nonparametric model with temperature as a covariate is compared to a model without temperature and to the original precipitation rates using a constant Z - R relationship. In addition, precipitation rates using separate rain $(Z - R)$ or snow $(Z_e - S)$ relationships are back calculated from the original precipitation rates and are compared to the nonparametric estimates. Further, we investigate if improvements in precipitation estimates varies with temperature ranges and if the method is dependent on the precipitation intensities.

## 2 Materials and methods

### 2.1 Study area

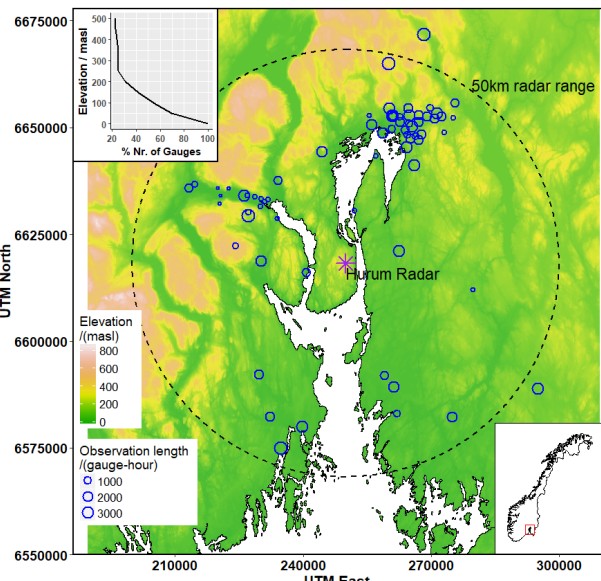

**Figure 1.** Precipitation gauge locations (blue circles) and length of the observations at each precipitation gauge location (size of the circles) and radar station (purple star mark) overlaid on topography of the study area, Oslo region of Norway. Hypsometric distribution (cumulative percentage of gauges below the specified elevation) of the gauges is on the top left corner.

The proposed nonparametric predictive model using radar precipitation rate and air temperature as covariates was tested on radar data over the Oslo region in Norway. The radar data used in the current research is an hourly radar precipitation rate product generated from the national weather radar network of Norway. The present study area is limited to the 50 km radius of radar range from Hurum radar station as shown in Fig. 1. The Hurum radar is located at 59.63° N latitude and 10.56° E

longitude and it is about 30 km from Oslo, the capital city of Norway and it has been in operation since November 2010. Data for the period from January 2011 to May 2015 were used for this study.

## 2.2 Data

The Norwegian Meteorological Institute (met.no) operates nine C-band Doppler weather radar installations which covers the entire land surface of Norway. The sensitive C-band installations with smaller wavelengths (4 - 8 cm) are placed in the Nordic to detect snowfall and clear air echoes (Koistinen et al., 2004). The wave length of the Hurum radar is 5.319 cm. The Norwegian radar network scans the atmosphere with a 7.5 minute temporal resolution; however, the temporal resolution was 15 minutes until June 2013. The met.no processes the raw radar volume scan from the radar stations. The data goes through extensive quality control and data transformations before the radar products are distributed to end users (Elo, 2012). The met.no performs a routine that removes clutter and other noise (non-meteorological echo) from the radar scan first. Then it reconstructs the gap in the data caused by clutter. The processing algorithm segments the volumetric radar reflectivity data as convective or stratiform precipitation type and it computes the Vertical Profile of Reflectivity (VPR) depending on precipitation types. VPRs of convective and stratiform precipitation types are distinctly different (Abdella, 2016; Chumchean et al., 2008). Bright band effect and non-uniform vertical profile of reflectivity are major sources of uncertainties in radar precipitation estimation in high latitude regions (Abdella, 2016; Joss et al., 1990; Koistinen et al., 2004; Koistinen and Pohjola, 2014). The radar data are corrected for bright band effects that appear in the VPR.

After the processing, the met.no generates and distributes various radar products. One of the radar precipitation rate products available for the public to use in hydrological applications is the Surface Rainfall Intensity (SRI). The SRI product uses the lowest Plan Position Indicator (PPI) and projects the aloft reflectivity data down to a reference height (1 km) near to the ground. The projection method is known as VPR correction that takes the vertical variability of reflectivity and bright band effect into account (Elo, 2012). The VPR corrected reflectivity is transformed from polar to Cartesian coordinate system with $1 \, \mathrm{km} \times 1 \, \mathrm{km}$ spatial resolution and the mosaic of nine weather radar imageries is merged to single SRI product covering the entire Norway. Finally, the reflectivity is converted to precipitation rate by using parametric Z - R relationship ($Z = 200R^{1.6}$) derived by Marshall and Palmer (1948) and the precipitation rate is accumulated to the temporal resolution desired (hourly in this case). The accumulated hourly SRI product was used in this study. It can be noted that the met.no uses the single Z - R relationship (Marshall-Palmer for rain) for all seasons throughout the year.

Within the study area, there are 68 precipitation gauges with available hourly precipitation data for the study. The gauges in the study site consists of Geonor weighing gauges and tipping bucket gauges. Both types are with an Alter wind shield. The met.no manages the calibration of gauges and takes necessary measures to reduce the uncertainty that arises when solid precipitation has to be measured. Further, data from the gauges are gone through routine quality control before being released to the public. However, the met.no does not do wind induced undercatch correction for the precipitation data.

The precipitation gauges' locations used in the study are shown in Fig. 1 overlaid on the topography of the study area. As shown in Fig. 1, precipitation gauges are not evenly distributed. The urban areas are densely gauged (Nearly 0.25 gauges/$\mathrm{km}^2$ near Oslo and approximately 0.1 gauges/$\mathrm{km}^2$ near other major cities) and rest of the area is sparsely gauged with hourly

**Table 1.** Different datasets used in the study and their source and spatial distribution.

| Description | Gauge precipitation | Radar precipitation | Air temperature | Wind speed | Relative humidity |
|---|---|---|---|---|---|
| Spatial Distribution | Gauge locations | Gridded (1 km x 1 km) | Gridded (1 km x 1 km) | Gridded (1 km x 1 km) | Gauge locations |
| Data Source | Gauge | Radar | Gauge (interpolated) | NORA10 and AROME | Gauge |

observation. Further, the precipitation data from precipitation gauges come with varying length because some gauges have been in operation since 2013 or later and some gauges have a number of missing values during their operation. Some of the gauging stations are equipped with hourly temperature and other meteorological measurements (including wind speed and relative humidity).

In addition to precipitation and air temperature data, wind speed and relative humidity data were also required for this study. The wind speed was used for undercatch correction of precipitation gauges and relative humidity was used in the precipitation phase computation. Table 1 describes the datasets used in the study and the source and the spatial distribution of each dataset.

The hourly gridded (1 km x 1 km) air temperature and wind speed datasets were generated by met.no. Lussana et al. (2016) spatially interpolated the past observed air temperature records from meteorological stations to develop the hourly gridded
temperature dataset for Norway using Optimal Interpolation. In this three-dimensional spatial interpolation, the elevation of each grid point was obtained from a high-resolution digital elevation model and the real elevation of stations were used. The resulting interpolated air temperature is on the regular grid which is 2 m above the ground terrain elevation. For further details of the interpolation method, readers are referred to met.no's report by Lussana et al. (2016).

The met.no derived an hourly gridded wind speed dataset by statistical downscaling from the 10 km numerical dataset,
"NORA10" (Reistad et al., 2011) combined with data from the "AROME" 2.5 km numerical dataset (Müller et al., 2017) using a local quantile regression (Lussana 2018, personal communication). The 1 km $\times$ 1 km grid of the wind speed data is the same as for temperature. Even though, radar precipitation rates and air temperature data are available from January 2011 to date, the unavailability of wind speed data for undercatch correction after 2015 limited the study period to four years (January 2011 - May 2015).

Hourly measured relative humidity data are available at 25-gauge locations within the study area. Spatial variation of relative humidity is relatively small within 50 - 100 km distances (Beek, 1991). It can be noted that nearest gauge with relative humidity measurement is less than 50 km for most gauges in this study and data from the nearest gauge was used for gauge locations without humidity measurements.

The datasets were downloaded and prepared for the study as follows. A spatial subset of hourly radar precipitation rate,
air temperature and wind speed data with 1 km $\times$ 1 km spatial resolution for the study area was downloaded from met.no's "thredds" server (http://thredds.met.no/). The data are in netCDF file format in UTM33N projection. The hourly precipitation measurements from 68 precipitation gauges and relative humidity measurements of 25 gauges were downloaded from met.no's web portal for accessing meteorological data for Norway, "eKlima" (http://eklima.met.no).

As precipitation gauge locations and radar precipitation rate grids are in the same UTM33N coordinate system, they were
simply overlaid and the radar pixel of 1 km$^2$ overlapping each precipitation gauge was located. One location near Oslo has

three precipitation gauges within a $1 \text{ km} \times 1 \text{ km}$ pixel. Except for that, all pixels consist of a single gauge. The pixel value (radar precipitation rate) for each hour was extracted and continuous hourly time series of radar precipitation rates for all gauges were generated. Similarly, time series of air temperature and wind speed at gauge locations were derived from the gridded temperature and wind speed data respectively.

The precipitation intensities in the study area is relatively low. An analysis of statistical properties of precipitation rates in mid Norway showed that intensities less than $1.76 \text{ mm h}^{-1}$ contributes to 50 % of the total precipitation volume while less than $6 \text{ mm h}^{-1}$ contributes to 88 % (Engeland et al., 2014). Further, the same study found that precipitation intensities below $0.1 \text{ mm h}^{-1}$ contributes little to the total precipitation and might be treated as zero precipitation. In addition, an analysis of the data used in this study showed that intensities between $0.05 \text{ mm h}^{-1}$ and $0.1 \text{ mm h}^{-1}$ are nearly 10 % of the total data above

$0.05 \text{ mm h}^{-1}$. Timesteps with gauge precipitation or radar precipitation rate less than $0.1 \text{ mm h}^{-1}$ were therefore removed in this study. Finally, an observed dataset of hourly gauge precipitation and corresponding radar precipitation rate and air temperature for those hourly timesteps were prepared for all precipitation gauge locations. The length of the dataset (number of gauge-hours) at each gauge location used in this study is shown with the size of the circles in Fig. 1. It can be noted that nearly 103000 total gauge-hours were used for the study.

Solid precipitation exhibits significant under-catch in windy conditions. Consideration of undercatch is more important in high latitude and mountainous regions due to high wind conditions. A Field study in Norway showed that precipitation gauges, even with wind shield, catch 80 % of true precipitation at wind speeds of $2 \text{ m s}^{-1}$, 40 % at $5 \text{ m s}^{-1}$, and only 20 % at $7 \text{ m s}^{-1}$ for solid precipitation at temperatures equal or below $-2° C$ (Wolff et al., 2015). As this study uses gauge observation as a ground observed truth, corrected gauge observation is required for a reliable outcome from the investigation.

We corrected gauge precipitation for wind induced undercatch by using the Nordic precipitation correction model (Førland et al., 1996). The Nordic model classifies the precipitation phase using air temperature and uses different equations for solid and liquid precipitation and a average value of the two equations is used for mixed precipitation. The correction equations use wind speed and air temperature at each gauge location. To verify whether the gridded wind speed data used in this study would provide a realistic correction, we compared it with the corrected precipitation using measured wind speed at 15 gauge locations.

It was found that correlation between the corrected precipitation by using measured wind speed data (15-gauge locations) and gridded data are over 0.97 for all 15 gauge locations. Based on the undercatch computations in this study, the mean correction factor of hourly precipitation (corrected precipitation/observed precipitation) is 1.61 for solid and 1.14 for liquid precipitation while median are 1.53 and 1.11 for solid and liquid precipitation respectively.

## 2.3    Methodology

### 2.3.1    Radar precipitation estimation

The proposed radar precipitation estimation algorithm consists of two steps. The first step quantifies the partial dependence of precipitation on radar precipitation rate and incident air temperature. The second step then uses the identified predictors

in a non-parametric setting to estimate the precipitation response. Gauge precipitation is used as a ground reference or true precipitation in this study.

The conditional estimation of precipitation using the two covariates can be described as follows:

$$R_{est}(t)|[R(t),T(t)] \tag{1}$$

Here, $(R_{est}(t))$ is the estimated ground precipitation from a given pair of radar precipitation rate $(R(t))$ and incident air temperature $(T(t))$ values at a given geographical location in the two-dimensional space (x, y) and time, $t$.

The conditional estimation in Eq. (1) uses two covariates, in contrast to Hasan et al. (2016a, b) where a nonparametric kernel regression estimator using a single covariate (R(t)) was adopted. Readers are referred to (Mehrotra and Sharma, 2006; Sharma and Mehrotra, 2014; Sharma et al., 2016) for further details on the nonparametric modelling framework used in this work. This study uses the k-nearest neighbour (k-nn) regression estimator as the nonparametric predictive model. This model can be expressed as:

$$E\Big(R_{est}(t)|[R(t),T(t)]\Big) = \sum_{k=1}^{K} \frac{\dfrac{g_k}{k}}{\sum_{j=1}^{K}\dfrac{1}{j}} \tag{2}$$

Where $k$ denotes the number of observed pairs of radar precipitation rate and air temperature considered "similar" to the current conditioning vector $[R,T]$. Similarity here is defined on the basis of a weighted Euclidean distance that is further explained below. $E(.)$ denotes the expectation operator, in the absence of which the uncertainty about the expected value can be computed. The term $g_k$ represents the observed gauge precipitation corresponding to $k^{th}$ neighbour of the conditioning vector. $K$ is a maximum number of neighbours permissible and it is an important parameter in the k-nearest neighbour method. In the present study, K is taken as equal to the square root of the sample size as suggested by Lall and Sharma (1996).

The order of each neighbour is ascertained based on a weighted Euclidean distance metric, written as:

$$\xi_i^2 = \left(\frac{\beta_R(R-r_i)}{s_R}\right)^2 + \left(\frac{\beta_T(T-t_i)}{s_T}\right)^2 \tag{3}$$

Here, $\xi_i$ is the distance of the conditioning vector $[R,T]$ to the $i^{th}$ data point $(r_i,t_i)$ in a two-dimensional space. $s_R$ and $s_T$ are sample standard deviations of the radar precipitation rate and temperature, and $\beta_R$ and $\beta_T$ are partial weights denoting the relative importance each conditioning variable has on the ensuing response respectively (Sharma and Mehrotra, 2014). The sample standard deviations are used to standardise the predictor variables to make them independent of their measurement scale, while the partial weights allow elimination of a predictor variable if not relevant to the prediction being made. Readers are referred to Sharma and Mehrotra (2014) for the informational theory rationale and partial informational correlation (PIC) that allows for the estimation of these partial weights, and the NPRED, R package ((Sharma et al., 2016), downloadable from http://www.hydrology.unsw.edu.au/download/software/npred) that enables their estimation for any sample data set.

### 2.3.2 Model evaluation criteria

A number of metrics have been used in literature to evaluate and compare the performance of models (Hasan et al., 2016a; Villarini et al., 2008). The root mean square error (RMSE) is commonly used as a performance measure and it provides the overall skill measure of a predictive model (Hasan et al., 2016a). We used primarily RMSE as a quality metric to evaluate the performance of the proposed model. Mean absolute error (MAE) and mean error (ME) were used as additional quality metrics. Definition of RMSE, MAE and ME can be found in the literature (e.g., Hasan et al., 2016a; Villarini et al., 2008).

### 2.3.3 Determination of phase

In order to assess the usefulness of the proposed approach, it was compared against an alternate approach where the precipitation phase was first ascertained, followed by the application of different Z-R relationships for snow and rain. For the classification of precipitation phase at gauge level, we adopted the method from Finnish Meteorological Institute which is used operationally in Finland for phase classification (Koistinen et al., 2004; Saltikoff et al., 2015):

$$P_{lp} = \frac{1}{1 + e^{22 - 2.7T - 0.2H}} \tag{4}$$

Here, $P_{lp}$ represents the probability of liquid precipitation, $T$ ($^{\circ}C$) the air temperature, and H (%) the relative humidity at a height of 2 m. If $P_{lp} < 0.2$, precipitation is considered as solid and if $P_{lp} > 0.8$, precipitation is considered as liquid. For the case of $0.2 \leq P_{lp} \leq 0.8$, precipitation is considered as mixed (Koistinen et al., 2004; Saltikoff et al., 2015).

## 3 Results

### 3.1 Partial weight of predictors

For each precipitation gauge location, we estimated the partial weights associated with radar precipitation rate and incident air temperature using the observed hourly radar precipitation rate and air temperature and the corresponding gauge precipitation data.

Figure 2 shows the histogram of partial weight of radar precipitation rate $(\beta_R)$ computed for the 68 precipitation gauge locations in the study area. It is noted that the summation of partial weights of radar precipitation rate $(\beta_R)$ and air temperature $(\beta_T)$ is scaled to 1. Hence, the partial weight associated with air temperature $(\beta_T)$ is equal to $1 - \beta_R$. Looking at Fig. 2, almost 87 % of the gauge locations resulted in non-zero partial weight for air temperature $(\beta_T > 0)$. In these locations, radar precipitation estimation partially depends on air temperature. It can be seen that partial weight of radar precipitation rate $(\beta_R)$ is equal to 1 for nearly 13 % of the gauge locations and the partial weight associated with air temperature $(\beta_T)$ is therefore zero. There, the bivariate problem collapsed into a univariate problem with radar precipitation rate as a single predictor.

Table 2 shows the summary statistics of computed partial weights among the precipitation gauge locations in the study area. It can be seen that the partial weight associated with air temperature is in the range of mean $+/- 0.1$ for more than 70 %

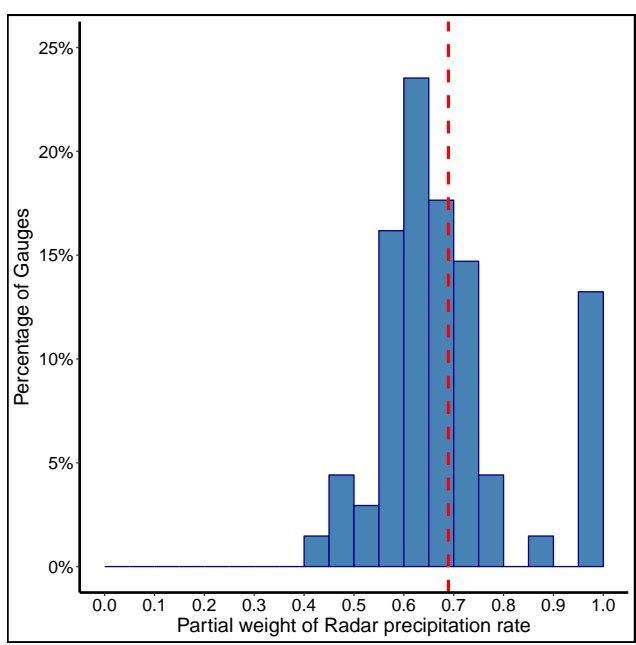

**Figure 2.** The Percentage of precipitation gauge locations against estimated partial weight of radar precipitation rate ($\beta_R$) at those gauge locations and the mean partial weight (red dash line) for gauge locations (68 gauges) in the study area. Partial weights provide a measure of relative importance of predictor variables on the response (refer Eq. (3)) and the summation of partial weights ($\beta_R + \beta_T$) is equal to 1.

**Table 2.** Summary statistics of computed partial weights for radar precipitation rate and air temperature in the study area.

| Partial Weight | Mean | 1st Quartile | 3rd Quartile | 15th Percentile | 85th Percentile |
|---|---|---|---|---|---|
| Radar precipitation rate ($\beta_R$) | 0.68 | 0.60 | 0.73 | 0.57 | 0.79 |
| Air temperature ($\beta_T$) | 0.32 | 0.40 | 0.27 | 0.43 | 0.21 |

of gauge locations. The gauge locations which resulted in associated partial weight for air temperature ($\beta_T > 0$) are spread throughout the study area. However, we have not found a clear pattern of spatial variation in the estimated partial weights.

### 3.2 Performance of k-nn prediction model

The k-nearest neighbour regression based estimator was used to estimate precipitation at each gauge location. The observed dataset and the computed partial weights of predictors were used with the NPRED k-nn regression tool to specify the proposed model with radar precipitation rate and air temperature as two predictors (knn-RT). For comparison, a reference model using the k-nn regression estimator with radar precipitation rate as a single predictor variable (knn-R) was also developed.

We calculated the k-nn regression estimate of expected response by using the leave-one-out cross-validation (LOOCV) procedure, whereby leaving out one observed response value (gauge precipitation) from the regression and estimating the expected response value for that observed response. This ensures the modelled outcomes represent the results that will be obtained using a new or independent data set. The improvement in radar precipitation estimation with the use of air temperature
as an additional covariate is measured as a percentage reduction in RMSE compared to the reference model.

All the gauge locations with an associated partial weight of air temperature ($\beta_T > 0$) show an improvement in radar precipitation estimation. The mean improvement in RMSE is 9 % and the improvement is more than 5 % for 80 % of the gauge locations where $\beta_T$ is greater than zero. It can be noted that partial weight for each gauge location was calculated independently using the data from that specific location and then the RMSE was estimated by LOOCV estimated using the entire data at that
gauge location. However, a split sample test was done to verify the results, where two-thirds of the data were used to estimate partial weight and one-third of the data were used to estimate RMSE for each gauge location. The split sample test gave similar results.

We also examined the spatial cross-validation of computed partial weights. First, a single average partial weight was calculated by taking the arithmetic mean of the partial weights for all gauge locations presented in Fig. 2 and Table 2. This single
average value of partial weight (0.68, 0.32) was used with the predictive models to estimate radar precipitation and the improvement in RMSE was estimated. Then, for each gauge location, an average partial weight was calculated by leaving that gauge out and adopting the mean partial weight from five nearest gauges. The k-nn prediction model was again re-specified for each gauge location using the computed average partial weight of the 5 nearest gauges. The percentage improvement in RMSE obtained by this method showed a strong resemblance to the results with a single mean value of partial weight. It is possible,
therefore, that a regional or nearest neighbour average value of partial weight can be used for ungauged locations. As with the partial weight, the improvement in RMSE at gauge locations does not show any pattern of spatial variation.

Based on above examinations, the spatial variation of station specific partial weights can be discarded and a single average value adopted. Hence, in the results that follow, we use a single average partial weight computed for the study area. As shown in Table 2, the mean value of partial weight for the radar precipitation rate is 0.68 and 0.32 for air temperature. The proposed
k-nn regression prediction model with radar precipitation rate and air temperature as two predictors at each gauge location was specified with this single average partial weight.

Figure 3 shows the percentage improvement in RMSE for the proposed model with the single average partial weight of (0.68, 0.32) compared to the reference model. The precipitation gauges' locations are shown by circles and a filled discrete colour scale is used to show percentage improvement in RMSE. All the gauge locations show improvement in RMSE with the use
of temperature as an additional covariate compared to the reference model. Looking at Fig. 3, the majority of gauge locations have a green colour and the improvement is between 5 - 10 % at those locations. The mean value of improvement is 8.5 %. Over 80 % of the gauge locations in the study area show more than 5 % improvement in RMSE while nearly 15 % show more than 15.0 % improvement. As discussed earlier and as seen in Fig. 3, this study did not find any pattern of spatial variation in the results. However, the spatial plot shows the improvement in RMSE with the use of temperature as an additional predictor
is spread throughout the study area.

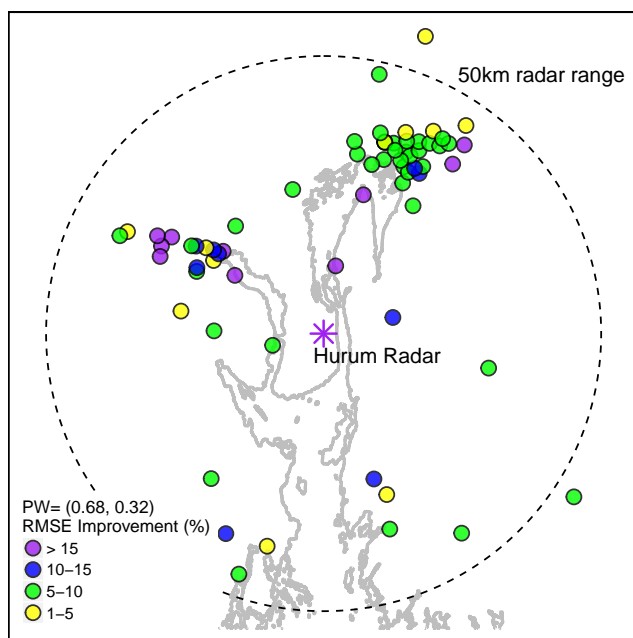

**Figure 3.** The percentage of improvement in RMSE at each gauge locations (colour scale) for predictive model with radar precipitation rate and air temperature as two predictors with the singe average partial weight ($\beta_R = 0.68$ and $\beta_T = 0.32$) compared to radar precipitation rate as a single predictor, overlaid on the coastline of the study area.

In addition to RMSE, we computed MAE and ME for the proposed model and the reference model at gauge locations. The above quality metrics were also computed for the original data of radar precipitation rates for comparison.

Figure 4 shows the summary of computed quality metrics for the two predictive models (knn-R and knn-RT) and the original data of radar precipitation rates (MP). A bar plot representing these three quality metrics at each individual gauge location is available in the supplementary material (Supplementary Fig. 1). Looking at Fig. 4, the mean error of the original data (MP) was negative for almost all gauge locations. This shows the under estimation of radar precipitation compared to precipitation measured by the gauges. Both nonparametric predictive models reduce the mean error considerably and bring it to near zero while they reduce the RMSE and MAE significantly for almost all gauge locations. It can be seen from Fig. 4 (a) and (b) that the predictive model with radar precipitation as a single predictor (knn-R) reduces the RMSE and MAE. The proposed predictive model with radar precipitation and air temperature as two predictors (knn-RT) further reduces both RMSE and MAE and improves the radar precipitation estimation for most of the gauge locations.

### 3.3 Performance for different threshold intensities

The study used the precipitation intensities of radar precipitation and gauge precipitation equal or above $0.1$ mmh$^{-1}$. As described in Sect. 2.2, precipitation intensities are relatively low in this region, consistent with intensities in cold climates. An

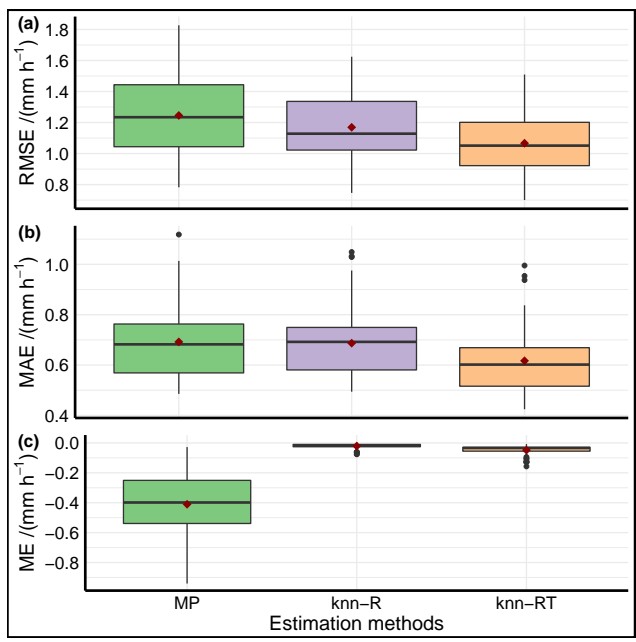

**Figure 4.** Box plot representing three quality metrics (RMSE, MAE and ME) estimated at gauge locations for the original data (MP) and for the two nonparametric models (knn-R and knn-RT). Here, knn-R denotes the nonparametric model with radar precipitation rate as a single predictor, while knn-RT denotes the nonparametric model with radar precipitation rate and air temperature as two predictors with fixed partial weight of (0.68, 0.32). The values outside $1.5 * IQR$ are represented by the whiskers.

analysis of the data used in this study showed that intensities are lower than $0.5 \text{ mm h}^{-1}$ for around 60 % of the observations and only 5 % of the data have either gauge or radar precipitation rates above $2.0 \text{ mm h}^{-1}$.

To investigate whether very low intensities dominate the results presented earlier, we tested our proposed model for a range of intensities for both gauge and radar precipitation. Figure 5 shows the box plot of RMSE values estimated at gauge locations for threshold intensities $0.1 \text{ mm h}^{-1}$, $0.5 \text{ mm h}^{-1}$ and $2.0 \text{ mm h}^{-1}$. Looking at Fig. 5, the improvement with the use of air temperature as an additional covariate is still seen over the intensity threshold. The results are statistically significant as the RMSE was estimated using leave one out cross validation (LOOCV) and are not impacted by the complexity of the model used.

### 3.4 Variation with Temperature Classes

For each gauge location, we also estimated partial weights for different temperature classes. The Partial Informational Correlation (PIC) and hence the partial weight was found to vary with temperature classes. For temperatures warmer than $10°$ C, most of the gauge locations were estimated as having zero partial weight for air temperature while those locations resulted in non-zero partial weight ($\beta_T > 0$) for temperatures colder than $10°$ C. It is therefore likely that radar precipitation estimation depends on air temperature mainly in colder temperatures.

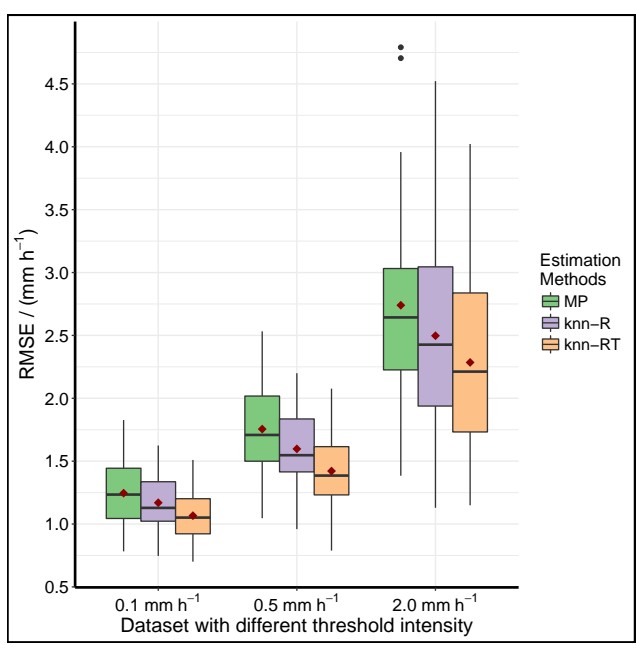

**Figure 5.** Box plot of $RMSE$ $(\mathrm{mm\,h^{-1}})$ values estimated at gauge locations for the original data (MP) and the two nonparametric models (knn-R and knn-RT) using data with intensities of radar precipitation rate and gauge precipitation greater than or equal $0.1\,\mathrm{mm\,h^{-1}}$, $0.5\,\mathrm{mm\,h^{-1}}$ and $2.0\,\mathrm{mm\,h^{-1}}$. Mean value of RMSE for each model by red diamond point. Here, knn-R - nonparametric model with radar precipitation rate as single predictor and knn-RT -nonparametric model with radar precipitation rate and air temperature as two predictors with the partial weight of (0.68, 0.32). The values outside $1.5 * IQR$ are represented by the whiskers.

Further, we estimated RMSE for the datasets with temperatures colder and warmer than $10°$ C for each gauge location using the proposed model with the average partial weight (0.68, 0.32) and the reference model. The proposed model reduces the RMSE significantly for temperatures colder than $10°$ C; however, the performance is nearly as same as the reference model for temperatures warmer than $10°$ C (Supplementary Fig. 2). This shows that the use of air temperature as an additional covariate
5  is most useful for the temperatures colder than $10°$ C.

### 3.5 Separate parametric equations for snow and rain as a benchmark

As discussed earlier, the switch between a snow and rain Z - R relation is fast becoming a standard for weather radar operations in cold climates. We compared the proposed nonparametric radar precipitation estimation model with radar precipitation estimation by using two different parametric Z - R relationships, one for snow and other for rain. In this study, we used
10  the radar snow equation of Finish Meteorological Institute $(Z_e = 100S^2)$ while keeping the Marshall and Palmer equation $(Z = 200R^{1.6})$ for rain.

The analysis reported so far in the paper is based on the accumulated hourly radar precipitation rate product available from met.no. The evaluation using separate parametric equations for snow and rain as a bench mark requires radar reflectivity data

to recompute radar precipitation rate using separate Z-R relationships for snow and rain. The reflectivity data used to produce the accumulated hourly radar precipitation rate (SRI product) used in the study are not stored in the production process, and therefore not available at met.no. As mentioned previously, the hourly product is based on corrected reflectivities with a time resolution of 15 minutes (before 2013) and 7.5 minutes (after 2013). These are then accumulated to the final hourly product.

However, the Plan Position Indicator (PPI) of the lowest elevation beam from Hurum radar is available from met.no.

To back calculate reflectivites with original short time resolution based on the available hourly radar precipitation rate, it was assumed that the precipitation intensity distribution in each hour is the same for both the SRI and the PPI product, and that the hourly precipitation rates (SRI) therefore could be distributed within the hour using the intensity distribution of the PPI data. This procedure then gives us a series of precipitation rates with a time resolution of either 15 or 7.5 minutes depending

on the year. The estimated precipitation rates were then converted to reflectivities using an inverse of the Marshall and Palmer equation ($R = (Z/200)^{1/1.6}$).

We estimated the probability of liquid precipitation ($P_{lp}$) using Eq. (4) and applied two different Z - R relationships to compute the precipitation rate according to the precipitation phase. Hourly air temperature and relative humidity at each gauge location were used to estimate the probability of liquid precipitation ($P_{lp}$). Data were classified as solid or liquid or mixed

precipitation using the computed hourly value of probability of liquid precipitation ($P_{lp}$). The back calculated reflectivity was converted to precipitation rates using the snow equation ($Z_e = 100S^2$) for solid phase and the rain equation ($Z = 200R^{1.6}$) for liquid phase. A weighted combination of solid and liquid was used for mixed precipitation by using the value of $P_{lp}$ as recommended by Koistinen et al. (2004); Saltikoff et al. (2015). The precipitation rates were then accumulated to hourly time resolution. The Precipitation rates estimated by the two equations as described above is denoted by FMIMP for the further

analysis.

For each gauge location, RMSE was calculated for the estimated radar precipitation rates by two equations (FMIMP). Here undercatch corrected gauge precipitation was used as a true observed value. RMSE of FMIMP is compared with the RMSE of original radar precipitation rates (MP) and the proposed nonparametric predictive model (knnRT). Figure 6 shows the box plot comparison of RMSE values in $\mathrm{mm\,h^{-1}}$ estimated at gauge locations for entire data and phase classes separately.

Looking at Fig. 6, the use of two equation (FMIMP) with the snow equation for solid and partially for mixed phase reduces the RMSE for solid and mixed precipitation phase classes and hence the RMSE of entire dataset compared to the original precipitation rates estimated by Marshall and Palmer equation (MP). The application of a different equation for snow reduces the phase dependent bias in the Norwegian radar precipitation estimation. The average reduction in RMSE at gauge locations is 6 % of RMSE value of the original radar precipitation rates. However, it can be seen in Fig. 6 that the use of different equations

for snow and rain does not reduce the RMSE to the level of the nonparametric approach (knn-RT). Comparing FMIMP and knn-RT, there is a further reduction of nearly 10 % in RMSE.

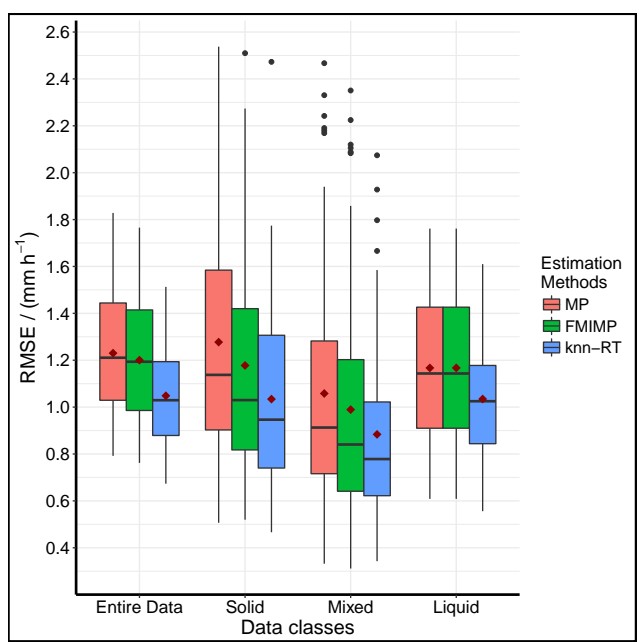

**Figure 6.** Box plot of comparison of $RMSE$ $(mm\ h^{-1})$ estimated at gauge locations for the original precipitation rates by Marshall and Palmer equation (MP) and precipitation rates estimated by different equation for snow and rain (FMIMP) and for the nonparametric model (knn-RT). RMSE values shown for entire data and separately for solid, mixed and liquid phase classes. Mean value of RMSE for each model by red diamond point. Here knn-RT - nonparametric model with radar precipitation rate and air temperature as two predictors with the partial weight of (0.68, 0.32). The values outside $1.5 * IQR$ are represented by the whiskers.

## 4 Discussion

In colder climates, the phase dependent uncertainties in radar precipitation estimation have hampered the extensive use of radar precipitation in hydrological applications (Berne and Krajewski, 2013; Saltikoff et al., 2015). To improve the quantitative radar precipitation estimates for hydrological applications, the study assessed the relevance of air temperature as an additional factor in the computation of radar precipitation in cold climates. In this paper, we show that using near surface air temperature as a second predictor variable in a nonparametric k-nearest neighbour (k-nn) method reduces the root mean squared error significantly compared to a k-nn model with radar precipitation rate as a single predictor and to the original hourly radar precipitation rates.

Despite phase dependent bias, accumulated radar precipitation rate products (e.g., met.no and OPERA) derived using a single Z-R relationship have been distributed to end users (Elo, 2012; Michelson et al., 2012). A key objective of the current study is to improve the hourly radar precipitation rates available to the public as a finished product (SRI product) from met.no that covers the entirety of Norway. However, the findings from this study can be helpful not only in Norway but also in a number of places where accumulated hourly product using a single Z - R relationship is applied. It can be noted that reflectivity

data (dBZ) could be used instead of radar precipitation rate in the methodology presented in the paper if such data are available as shown by Hasan et al. (2016a).

A nonparametric framework was used for the investigation posed in the paper. Earlier studies (Hasan et al., 2016a, b) reported that given the availability of large amount of radar data, nonparametric approaches produce more reliable radar rainfall estimates compared to a traditional parametric Z - R relationship. In these studies, the nonparametric model used the radar reflectivity as a single predictor. This is the first study to our knowledge that considered the air temperature as an additional covariate in the radar precipitation estimation and the approach provided a clear improvement in the estimation. However, the improvement was significant for temperatures colder than $10°$ C. This appears mostly due to the different phase of precipitation in colder temperatures (including the presence of hail).

Partial informational correlation (PIC) based partial weights were used first to assess the partial dependence of radar precipitation estimation on air temperature, and then the weights were used with the k-nearest neighbour (k-nn) model. A simple k-nn approach is to use an equal weight for predictor variables or weights estimated using a simple linear partial correlation. Mehrotra and Sharma (2006) argue that the approach of assuming both predictor variables are equally important can result in increased bias and predictive uncertainty. Moreover, earlier studies (Mehrotra and Sharma, 2006; Sharma and Mehrotra, 2014; Sharma et al., 2016) have shown that the estimated PIC and weights collapse to what would be estimated using a linear regression model if the system is linear. As the system here is nonlinear, the use of the PIC to estimate partial weights seems to be the best approach. The study used a single average partial weight for the study area. If needed, it can be possible to use gridded partial weight with the k-nn model. However, we found that the gain in RMSE is not significant for the effort of added complexity of gridding the partial weights.

Although the main focus of this paper is to investigate the benefit of using air temperature as an additional covariate in radar precipitation estimation, the results of the nonparametric method of radar precipitation estimation found are comparable with the results of Hasan et al. (2016a). They tested their kernel based nonparametric method of radar rainfall estimation (radar reflectivity as a single predictor) in Sydney, Australia and reported a 10 % improvement in RMSE compared to the traditional parametric Z - R relationship. In this study, the k-nearest neighbour nonparametric method with radar precipitation rate as a single predictor resulted in a mean reduction in RMSE of 6 %. The proposed bivariate k-nn model with air temperature as an additional predictor resulted in a mean reduction in RMSE of 14 % compared to the original radar precipitation rates data.

The near surface air temperature, also together with relative humidity or wet-bulb temperature, has been used to estimate the dominant phase of precipitation in the selection of Z - R relationships (Koistinen et al., 2004; Saltikoff et al., 2015, 2000). Fassnacht et al. (1999) and Fassnacht et al. (2001) reported the use of near surface air temperature to adjust the radar precipitation estimation and the benefit of the adjustment for hydrological applications. However, their approach was to use the temperature to estimate the probability of snow and use that information for the adjustment of radar precipitation. Further, the method was limited to mixed precipitation only, while the work presented here adjusts precipitation rate (it could be rain or snow or a mixture thereof) by using the k-nearest neighbour approach with near surface air temperature as a covariate.

The performance of the proposed k-nn method with temperature as a covariate was assessed primarily using a k-nn model without temperature and original radar precipitation rates derived by a single Z - R relationship as bench marks. As most cold

climate radar operations use two separate equations for snow and rain, the study compared the nonparametric estimates with the precipitation rates estimated by two equations. First, reflectivities were back calculated in order to apply two equations. For this, we used PPI precipitation rates to distribute the VPR corrected SRI precipitation rates by assuming both have same intensity distribution within each hour. While there is uncertainty in how accurately the redistributed intensity distribution of SRI represents the original distribution, this exercise at least used a possible realistic distribution. Secondly, it should be noted that the phase classification used in this evaluation is a model-based classification even though it is used operationally. The estimated phase can differ from actual observed phase at gauge level. Observations from disdrometers can provide a more accurate phase information at gauge level. Even if a few disdrometers were located within the study region, their representativeness in space and time would be limited (Saltikoff et al., 2015). Further, our phase classification is at gauge level, and represents near surface conditions. The phase of the precipitation can be different at the elevation where the radar measures the reflectivity.

Air temperature can be lapsed to the radar measurement height to estimate the phase of precipitation. Fassnacht et al. (1999, 2001) assumed the temperature lapse rate to be zero in their studies since winter lapse rates is often zero in mid latitude areas. For the Nordic region, Tveito and Førland (1999) showed that the vertical lapse rate varies with season and location. Further, Tveito and Førland (1999) found that local terrain conditions have greater influence in local temperature gradient during winter. Due to the occurrence of inversions, lapse rate can deviate substantially from the standard ($-6.5^\circ\,\mathrm{C\,km^{-1}}$) during the winter months and it can be as low as $-1.2^\circ\,\mathrm{C\,km^{-1}}$ (Tveito et al., 2000; Tveito and Førland, 1999). The estimated temperature at radar measurement height and hence the probability of liquid phase ($P_{lp}$) are therefore highly uncertain (Al-Sakka et al., 2013; Tveito et al., 2000). We, therefore, use the Finnish Meteorological Institute's operational method of near surface phase estimation to classify the precipitation as the method of choice for the evaluation as this method is both in operational use and developed for the Nordic area (Koistinen and Saltikoff, 1998; Gjertsen and Ødegaard, 2005; Saltikoff et al., 2015). The measurements of phase information at radar measurement height with the use of dual polarized radars can be a useful data source (Ryzhkov and Zrnic, 1998; Chandrasekar et al., 2013; Al-Sakka et al., 2013) for further investigation. However, many radars use a single polarity and moreover, even from dual polarised radars, data on phase information are not readily available to end users to help refine their estimation algorithms.

The study used the undercatch corrected gauged precipitation as a ground truth. We did a test on uncorrected gauge precipitation data (not corrected for wind induced undercatch) during an early phase of the study and found that air temperature as a covariate lead to improved RMSE in radar precipitation estimates also with uncorrected precipitation. It is often challenging to get reliable wind speed measurements for an operational real time radar precipitation estimation, and this finding implies that the method can also be used with uncorrected gauge precipitation to adjust the radar precipitation rates.

The improved precipitation rates obtained through the nonparametric estimation of radar precipitation can be a data source for hydrological applications. The spatial detail of the radar precipitation could solve issues related to precipitation representativity for hydrological modelling (Smith et al., 2004; Kirchner, 2009; Hailegeorgis et al., 2016). For many hydrological applications, short duration precipitation is needed and extending the study to sub-hourly time resolution and multiple radar bands (e.g., X band, S band) would be an interesting continuation to this work.

# 5  Conclusions

While parametric phase dependent Z-R relationships adjusted with gauged precipitation have been discussed extensively in the literature, this study extends current work with air temperature as a covariate in the radar precipitation adjustment, further presents a procedure whereby precipitation can be estimated in colder climates.

An improvement of 15 % in the root mean squared error was obtained using a simple nonparametric method with air temperature as an additional covariate. More than 80 % of the locations showed improvement when temperature was used in the nonparametric model. The improvement was independent of precipitation intensities. However, the temperature effect became insignificant when air temperature was warmer than 10° C.

Given the importance of weather radars as a means of precipitation measurement, and their ability to observe in remote

regions in a continuous setting, the above finding could be important for using radar precipitation data for hydrological applications especially in cold climates.

*Code and data availability.*  Radar precipitation rate data used in the study are available in the Norwegian Meteorological Institute's (met.no) thredds server (http://thredds.met.no/thredds/catalog/remotesensingradaraccr/catalog.html). Precipitation observations from precipitation gauges, other meteorological measurements (wind speed and relative humidity) and gauges' meta information can be obtained from met.no's web

portal "eKlima" (http://eklima.met.no). Access to the web portal is available upon request. Gridded observational hourly air temperature data and gridded wind speed data are available in the met.no's thredds server (http://thredds.met.no/thredds/catalog.html). NPRED programming tool, which is used for computation in the study, is available as R package and it can be downloadable from the following link as follows: http://www.hydrology.unsw.edu.au/download/software/npred

*Competing interests.*  The authors declare that there are no competing interests.

*Acknowledgements.*  The authors gratefully acknowledge the Norwegian meteorological institute (met.no) for providing radar precipitation rate, gauge precipitation and meteorological data for this study. The authors would particularly like to thank Christoffer Artturi Elo and Cristian Lussana at met.no for assisting to get the radar precipitation rate and gridded meteorological data. A great appreciation goes to Water Research Centre, University of New South Wales (UNSW), Sydney, Australia for hosting the first author for research practicum. The authors acknowledge the Norwegian Research Council and Norconsult for funding this research work under the Industrial Ph.D. scheme

(Project No.: 255852/O30).

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
