# Peer review of "Estimating Radar Precipitation in Cold Climates: The role of Air Temperature within a Nonparametric Framework"

_Hydrology and Earth System Sciences, 2018_

## Referee Comment (RC1) · S.R. Fassnacht (Referee) · 13 Jul 2018

review of HESS-2018-0351 "Estimating Radar Precipitation in Cold Climates: The role of Air Temperature within a Nonparametric Framework," which is essentially a re-review of hess-2017-662 "Should radar precipitation depend on incident air temperature? A new estimation algorithm for cold climates." Since this is a new submission, I will review it as such.

General Comments:

This is an interesting paper that should give us some improved insight into using

weather radar to estimate rain and snow fall. This is very relevant in higher latitudes and in mountain environments were snow is important. To date, there has been limited work in using weather radar for snowfall estimation in a hydrological context. The methods presented herein could be used in many locations. However, the writing is unclear and I got lost at times. I suggest that the authors revisit their objectives and ensure that the paper addresses these. Also, the Discussion is essentially missing as the work is not put into context of the few other relevant studies. Below I outline restructuring and a problem with the Methods/Data.

Equation 4 uses air temperature and relative humidity to estimate the phase of the precipitation from Koistinen et al. (2004) and used by Saltikoff et al. (2015) for Finland. However, air temperature at the gauge is used, and this is not correct. Fassnacht et al. (1999; 2001) lapsed the air temperature up to the radar measurement height. There can be 5 to 10 degrees Celsius difference between the temperature at the height (2 m above the ground) and the radar height (1 km stated on page 8 line 15). At minimum this should be discussed?

What about a split sample approach of calibration and evaluation for the partial weights and k-nn approach? Also, the partial weight for radar precipitation was shown to vary from 0.4 to 1 (Figure 2), so why was a single average (mean) used in the k-nn prediction model. Is this approach not robust enough to have a different partial weight, or perhaps a gridded partial weight? It is stated that there is no spatial pattern in the partial weights, but an interpolated residual type approach could be used (e.g., Fassnacht et al., 2003 among others).

The paper does need restructuring and rewriting. At present I get lost in where I am in the text, regardless of the "foreshadowing" sentences that appear at the end of various sections. 1) At the end of the Introduction, the paper should tell the reader specific objectives that were investigated or research questions that were answered.

2) Some of the material in the Background is repeated from the Introduction. For example, the three paragraphs in section 2.1 (Radar precipitation estimation in cold climates) mostly in the Introduction. Either reduce the Introduction or merge the Background with the Introduction to remove the repetition. I suggest the latter and to consider adding sections to the Introduction (e.g., 1.1. Weather radar use for hydrology, 1.2. Radar precipitation estimation in cold climates, 1.3. Nonparametric Radar rainfall estimates).

3) There are methods presented in the Study Area and Data section. These two sections 3. Methods and 4. Study Area and Data need to be revisited to put all the methods together. I suggest a brief section first on Study Area, then a section on Data and Methods, describing the data first, then the methods used.

4) The Results and Discussion are combined and the Discussion is thus limited. I recommend that the Results and Discussion sections should be presented separately, or that the Discussion be much more in depth. There are only three citations in the entire Results and Discussion section, while numerous useful citations are presented in the Introduction and Background sections. There is no Discussion that put this work into context; the Results and Discussion only presents how do these results compare to the findings of Fassnacht et al. (1999), Koistinen et al. (2004) and Saltikoff et al. (2015).

5) At the end of the Summary and Conclusions, it is stated that "[w]hile this study uses data for one weather radar in arriving at its conclusions, preliminary analysis suggests the problems noted here to be generic." If there are additional "preliminary" results from some should be presented. This statement is important but is hanging.

6) In various locations throughout the text, sentences are added that foreshadow the next or subsequent sections. These are not necessary and should be removed.

The meshing of meteorological data with the radar data on a 1 x 1 km grid, including the interpolation of the station data is confusing. This is the four paragraphs on page 8 line 23 through page 9 line 21. This section needs to be rewritten, as it is unclear what is done. Perhaps a table could be added that describes the four datasets (T, RH,

wind, and radar). From the text, I assume that temperature and RH data have been gridded at a 1-km resolution from the station data: T using the Optimal Interpolation in a Bayesian setting (Lussana et al., 2016) and RH using the nearest neighbor. What is the Lussana et al. (2016) method? While wind data are available, they are downscaled from a 10 km numerical model dataset. What numerical model dataset is used? This section needs to provide more details - it does not have to be much longer, the methods just need to be clarified. Also, the gauge precipitation (Pgauge) data are used as point measurements meshed with the gridded data; this would also be included in the aforementioned table. These data are revisited in section 4.1.

The font size is to small in most figures and the text is often grey. This makes the figures difficult to read. his should be addressed throughout.

Specific Comments:

Page 1, line 1: I suggest saying "In colder climates ..."

p1, l1 and l2, p2 l1, etc.: be consistent with "form of precipitation" and "state of precipitation." I suggest calling it "phase of precipitation" throughout the text.

p1, l5: "estimate" or "adjust"?

p1, l11: usually "catch error" is called "undercatch"

p1, l15: do you mean gauge air "temperature" or temperature at the radar measurement height?

p1 l15 and subsequently: to be more specific, use "warmer" than instead of "above" when referring to air temperatures. "Above" implies an altitude above the ground, which is typically associated with a colder air temperature. p2, l25: use "colder than" instead of "below," etc.

p1, l15-16: the end of the sentence "which indicates that the partial dependence of precipitation on air temperature is most important for colder climates alone" is unclear.

[Figure]

Please reword.

p1, l22: should "2010b" be "2010a?" Check this, as (Villarini and Krajewski, 2010a) has not appeared yet.

p2, l13: why "Conventionally?" Use another word so that the reader does not confuse radar types, such as "the original way" (i.e., conventional), Doppler, dual-polar, multi-wavelength.

p2, l18: since Canada is mentioned here (Crozier et al., 1991) could be add to the citation list on line 20

p2, l25: add an "s" to "quarter"

p3, l2: "different temperatures cause different shapes of crystals." For solid precipitation, i.e., snow, the degree of super-saturation also affects the crystal shape.

p3, l5: what is meant by "multiple snow types?" Does this imply shapes? If so, state this explicitly.

p3, l8: in many cases the correlation between probability of snow and temperature is an "'S' shaped structure," (see Fassnacht et al., 2001 for a summary illustration), but a simpler linear relation has also been used (e.g., Fassnacht et al., 2013).

p3, l9-10: be specific with "[t]he dielectric property of solid particles (ice) is very different from liquid particles (water)." "Very different" is vague.

p3, l12: reverse the order of the Hasan et al. (2016) references. You present 2016b before 2016a.

p3, l13: change the word "Historical"

p3, l25-29: delete the sentences in the rest of the paragraph starting with "[t]he rest of the paper is structured as follows." You do not need to tell what the sections of the paper are, that reads like the end of a thesis. Instead, give us specific objectives to

investigate or research questions that are answered.

p3, l28: The results should be presented, then there should be a separate Discussion section.

p4, l27: there has also been some work on phase discrimination using multiple radar wavelengths (e.g., Al-Sakka et al., 2013).

p4 l29 to p5 l3: This paragraph can be reduced to 1-2 sentences, as this information is generally known.

p5, l12-13: please reconsider "nonparametric approaches ... weakness is that the method is sensitive to outliers." I am not sure that this is correct. Parametric approaches tend to be sensitive to outliers.

p5, l18-20: these two foreshadowing equations are not necessary.

p7, l4: "classification of precipitation phase at gauge level" is good, but don't we need the phase of precipitation at the radar height to select the appropriate radar Z-R equation? Although this is what Koistinen et al. (2004) and Saltikoff et al. (2015), it doesn't necessarily make it correct.

Figure 1: a) I assume that the "length of the observations" is the number of hours with precipitation? b) I also assume that the hypsometry curve is cumulative % of stations below the specified elevation. Please be specific. c) the font size is small and difficult to read. Enlarge and also don't use grey. d) are the red names local cities? Are they important? If so, move them so they are legible.

p7, l12-13: What is the "accumulated hourly radar precipitation rate product?" Is this accumulated from sub-hourly to yield an hourly total, or is the hourly data added?

p7, l14: tell us how many gauges in the "a relatively dense network of precipitation gauges."

p8, l26: instead of "are" use past tense through the methods.

p8, l34: reword the last sentence "However, we used data from all available precipitation gauges for this study." Perhaps state something about the total number of gauge hours of data used (likely in the order of 100,000 gauge-hours).

p9, l2: provide a sources for the "gridded hourly temperature and wind speed dataset"

p9, l6: delete "[m]ore details on the procedure adopted for catch correction are provided in the next sub-section."

p9, l10: change "resulted" to "resulting"

p9, l29: how little is "intensities below 0.1 mmh-1 contributes little?"

p10, l2-8: this is background. It could be moved to earlier in the text, as this is the methods/data section. Tell us what was done. This sentence could also be deleted.

p10, l4: the word "Nordic" is not necessary here, as it could also be relevant in southern environments

p10, l4-5: the end of the sentence is redundant "due to large catch errors for snow." In could state that it is "due to high wind conditions." It is wind that causes undercatch, not "large catch errors"

p10, l7: "Wolff et al., 2015" is not in the citation list

p10, l9 or previous: what type of precipitation gauge and what type of shield are used? This is very important information to assess the degree of undercatch and the error associated with the undercatch correction.

p10, l9-17: throughout this paragraph it is "undercatch" correction. This should be consistent, as there are other errors.

p10, l13-14: "It was found that correlation between the corrected precipitation by using measured wind speed data (15-gauge locations) and gridded data are over 0.97..." Does this mean the correlation undercatch correction using gauge wind speed versus

the downscaled gridded wind speed?

p10, l14: there are only 15-gauge locations with wind speed measurements. How are the other 53 precipitation gauges corrected for undercatch? From my comment above (p10, l13-14), I assume that the downscaled gridded wind speed was used to correct for undercatch at all 68 precipitation gauges. This is not clear.

p10, l19-22: delete this paragraph. We know what you are going to do Figure 2: add tick marks to the y-axis.

p10, l23 through p11: Are the "Partial weight of predictors" constant over time, i.e., is there a specific value for station that does not change?

Table 1: As this is a summary of Figure 2, this table could be converted to two horizontal box and whisker plots on Figure 1. The reader doesn't really care about the specific numbers, just the range.

p12, l1 and elsewhere: the word "prediction" implies that this is for the future. I suggest using "estimation" throughout.

Figure 3: the caption is confusing; break into two sentences. Also, are these "length of the data (circle size)" the same as in Figure 1? If so, then don't display again here, unless this is relevant later?

p13, l4-7: these three sentence could be reduced to a small histogram of RMSE reductions that is added to figure 3.

p12 to 15: why was a single average (mean) partial weight (beta P = 0.68) used in the k-nn prediction model, when it was shown (Figure 2 and Table 1) that the partial weight varies from 0.4 to 1?

p15 section 5.3: all this text except for the last sentence is background or methods and should be moved to an appropriate location earlier in the paper.

p15, l15: does this "still significant" has a statistical meaning? If so, explain how. If not,

don't use the word significant.

p15, section 5.4: what is the range of the "Temperature Classes?" You only discuss T > 10C. What about T < 10C? The point of this section is unclear.

Figure 6: what are the dots above the solid and mixed phase?

p18, section 5.6: these statements seem to be hanging. Can you present specifics?

p18, l5: what does "the use of temperature as an additional predictor variable is having consistent impact" mean? The words "consistent impact" are not clear

References

Al-Sakka, H., Boumahmoud, A.-A., Fradon, B., Frasier, S.J., Tabary, P.: A new fuzzy logic hydrometeor classification scheme applied to the French X-, C-, and S-band polarimetric radars, J. Appl. Meteor. Climatol., 52, 2328-2344, 2013.

Crozier, C.L., Joe, P.I., Scott, J.W., Herscotvitch, H.N., Nichols, T.R.: The King City operational Doppler radar: development, all season applications and forecasting, Atmosphere-Ocean, 29, 479-516, 1991.

Fassnacht, S.R., Dressler, K.A., Bales, R.C.: Snow water equivalent interpolation for the Colorado River Basin from snow telemetry (SNOTEL) data, Water Resources Research, 39(8), 1208, 2003.

Fassnacht, S.R., Venable, N.B.H., Khishigbayar, J., Cherry, M.L.: The Probability of Precipitation as Snow Derived from Daily Air Temperature for High Elevation Areas of Colorado, United States, Cold and Mountain Region Hydrological Systems Under Climate Change: Towards Improved Projections (Proceedings of symposium H02, IAHS-IAPSO-IASPEI Assembly, Gothenburg, Sweden, July 2013) IAHS, 360, 65-70, 2013.

---

## Referee Comment (RC2) · Anonymous Referee #2 · 26 Jul 2018

General comments:

The authors apply the non-parametric k - nearest neighbour method (k-nn) to estimate radar precipitation from gridded surface observations of rainfall and temperature for the Oslo region in Norway. They show that utilising temperature as second predictor variable reduces the root mean squared error significantly compared to a k-nn model without temperature and com-pared to the original procedure using a constant Z-R-relationship or separate snow/rain Z-R relationships.

The application of this method for radar rainfall estimation including temperature is novel and of interest not only for readers living in regions with colder climates. The

research is done systematically and quite carefully. The paper is written well and clear in structure. However, there are three major points and some minor things which need attention before the paper can be published. One main point are the lengthy introduction and background sections which could be shortened. A second important point concerns the method to estimate the partial weights. It becomes not clear, that this method is really providing optimal weights. And, third, there seems to be an issue with the back-calculation of Z using the inverse Z-R relationship on a different time resolution as for the original forward calculation. Detailed information about this and the minor things are given below.

Detailed comments:

1. Sections1 and 2: Both sections together cover almost 4 pages and represent the introduction with the state of the art. This is quite lengthy. The introduction is very general; the background is more focussed on the topic at hand. I would suggest to shorten these parts especially the introduction significantly and may be use the background as introduction.

2. Eq. 1: As predictor R(t) is used. Why not using Z(t) as predictor? For R(t) already a (wrong) Z-R-relationship has been applied, introducing great uncertainty. If a linear re-lationship is required a log-log transformation of Z(t) and Rest(t) could be applied be-forehand. This needs at least to be discussed.

3. Fig. 1: The units for observation length and elevation are missing. Also the text of the legend is tiny and nard to read.

4. Section 5.1: It is not clear if the estimation of the partial weights using partial information correlation (PIC) is really beneficial or even optimal. In order to prove the merit of PIC I would suggest to test two additional cases a) equal weights for P and T and b) using simple linear partial correlations. The performance for the latter two cases measured by RMSE should be worse than by PIC weighing.

5. Fig. 4: This bar plot is not easy to read. I would suggest to use box-whisker plots in-stead.

6. Page 16, line 1: The back-calculation of Z from R using a non-linear relationship on hourly data gives an estimated average Z value for each hour. This estimate can be quite different from the observed average Z value if the rainfall distribution within the hour is not unique. In the forward calculation the Z-R relationship is applied on 7.5 min Z values to calculate 7.5 minute rainfall intensities. Because of the non-linearity of the Z-R relationship a simple back calculation on a different time step than the one the original calculation was applied is not possible. For non-linear functions f is $E[f(x)] <> f[E(x)]$.

---

## Referee Comment (RC3) · G. Ravazzani (Referee) · 21 Aug 2018

General comments:

In this paper, a non-parametric method is applied to estimate radar precipitation considering both rainfall and temperature. The use of radar for precipitation estimation is an interesting topic. Many papers have been presented about this topic, but the specific problem authors deal in this paper is how to assess solid precipitation in cold regions. The solution they propose is of interest for cold climates in northern Europe, of course, but I suppose it could be extended to other areas where solid precipitation occurs.

[Figure]

Specific comments:

Authors used 68 rain gauges in this study that are clustered around urban areas. Do authors think that this uneven distribution may affect results? In other terms, is the location of raingauges relevant for the application of the proposed procedure?

P 9 L 14 "The gridded hourly wind speed datasets are derived from a statistical downscaling of a 10 km numerical model dataset onto a 1 km grid". Did authors verify how the method is sensible to the specific realization of the statistical downscaling?

Authors apply correction to gauge precipitation to consider wind induced underestimation. Gauge precipitation is affected by several sources of uncertainty. Wind is of course relevant, but another systematic error is related to the calibration of raingages that causes underestimation for high rainfall intensity and overestimation for low rainfall intensity. Further uncertainty arises when solid precipitation has to be measured. How did authors deal with these errors? Are they already managed by the meteorological institute?

Section 5.6 is very short compared to the rest of the paper and I did not fully understand what is the intention of authors. I think they should better explain this part or remove it.

Technical corrections:

P.4 L. 6 The Finnish Meteorological Institute

---

## Author Comment (AC1) · 27 Aug 2018

**Response to the review of hess-2018-0351**

**RC1: Responses to S.R. Fassnacht (Referee 1)**
The authors wish to thank the reviewer for his constructive comments and corrections to the discussion paper. In the following, we have responded to each of the comments from the reviewer. The comment from the reviewer (RC) is in italic font while the author comment (AC) and changes in the manuscript (CM) are in blue normal font.

*This is an interesting paper that should give us some improved insight into using weather radar to estimate rain and snow fall. This is very relevant in higher latitudes and in mountain environments were snow is important. To date, there has been limited work in using weather radar for snowfall estimation in a hydrological context. The methods presented herein could be used in many locations. However, the writing is unclear, and I got lost at times. I suggest that the authors revisit their objectives and ensure that the paper addresses these. Also, the Discussion is essentially missing as the work is not put into context of the few other relevant studies. Below I outline restructuring and a problem with the Methods/Data.*

*Equation 4 uses air temperature and relative humidity to estimate the phase of the precipitation from Koistinen et al. (2004) and used by Saltikoff et al. (2015) for Finland. However, air temperature at the gauge is used, and this is not correct. Fassnacht et al. (1999; 2001) lapsed the air temperature up to the radar measurement height. There can be 5 to 10 degrees Celsius difference between the temperature at the height (2 m above the ground) and the radar height (1 km stated on page 8 line 15). At minimum this should be discussed?*
AC: As it is mentioned, we adopted the operational method from the Finnish Meteorological Institute as presented by Koistinen et al. (2004) and Saltikoff et al. (2015). In their phase equation, near surface temperature (2m above ground) is defined and we did follow their method as defined in the papers. We do agree with the reviewer that air temperature at radar measurement height can be different from gauge height and hence the estimated phase can be different. This has already been discussed on p17, l12-13 "Further, our phase classification is at gauge level, and represents near surface conditions. The phase of the precipitation can be different at the elevation where the radar measures the reflectivity."

Air temperature can be lapsed to the radar measurement height to estimate the phase of precipitation. Fassnacht et al. (1999) and Fassnacht et al. (2001) assumed the temperature lapse rate to be zero in their studies as winter lapse rate is often zero in mid latitude areas. For the Nordic region, Tveito and Førland (1999) showed that vertical lapse rate varies with season and location. Further, Tveito et al. (2000) found that local terrain conditions have greater influence in local temperature gradient during winter. Due to the occurrence of inversions, lapse rate can deviate substantially from the standard (-6.5° C/km) during the winter months and it can be as low as -1.2° C/km (Tveito et al., 2000, Tveito and Førland, 1999). The estimated temperature at radar measurement height and hence the probability of liquid phase (Plp) are therefore highly uncertain. The measurements of phase information at radar measurement height with the use of dual polarized radars can be a useful data source for further investigation.

After receiving the reviewer's comment, we investigated the sensitivity of our results to the use of a lapse rate. The air temperature at the radar measurement height (1 km) was computed using a standard moist adiabatic lapse rate (-6.5° C/km) as used in Nordic meteorological studies (Tveito et al., 2000). We estimated the probability of liquid precipitation (Plp) in Eq. (4) by using the lapsed temperature at radar measurement height while assuming the relative humidity unchanged. The estimated Plp was used to classify the precipitation phase and repeated the work as presented in section 5.5. Our results (not included in the revised paper) showed that new classification did not improve RMSE compared to the use of near surface phase classification. We attribute this to the considerable uncertainty associated with the use

of the lapse rate as noted by others (Al-Sakka et al., 2013, Tveito et al., 2000). Further, equation (Eq. (4)) is developed and tested for surface phase classification and it incorporates relative humidity as a variable. We assumed relative humidity at the radar measurement height similar to the surface value. This is also a potential error source in the computation.

We, therefore want to keep the Finnish Meteorological Institute's operational method of surface phase estimation to classify the precipitation as the method of choice in the paper. This is both in operational use and developed for the Nordic area which gives us some confidence in the method.

CM: We discuss this issue in the revised manuscript and mention the lack of improvement when a default lapse rate is considered in our algorithm.

*What about a split sample approach of calibration and evaluation for the partial weights and k-nn approach? Also, the partial weight for radar precipitation was shown to vary from 0.4 to 1 (Figure 2), so why was a single average (mean) used in the k-nn prediction model. Is this approach not robust enough to have a different partial weight, or perhaps a gridded partial weight? It is stated that there is no spatial pattern in the partial weights, but an interpolated residual type approach could be used (e.g., Fassnacht et al., 2003 among others).*

AC: As described on p12, l15-17, a split sample test was done to verify the results obtained from the leave one out cross validation (LOOCV) approach and presented in the paper.

Partial weights did not show any spatial pattern that would allow us to generate an informed specification of the weights that could be applied over the study region. Further, the RMSE estimated at gauge locations with the single average partial weight for the study area (as we presented in this paper) showed a strong resemblance with the RMSE estimated by using the partial weight estimated from the five nearest gauges. Hence, we decided to use a single average partial weight to present in this paper.

As the reviewer mentioned, it is possible to use gridded partial weight or an interpolated residual type approach. However, we found that the gain in RMSE is not significant for the effort of using gridded or residual type partial weights. The added complexity of gridding the partial weights does not add significant information to the analysis and we therefore recommend using an average value in the computations for this study region.

*The paper does need restructuring and rewriting. At present I get lost in where I am in the text, regardless of the "foreshadowing" sentences that appear at the end of various sections. 1) At the end of the Introduction, the paper should tell the reader specific objectives that were investigated or research questions that were answered.*

AC / CM: As per the reviewer's suggestions we have added specific objectives at the end of the introduction along with stating them on p3, l22-25. These are stated here for brevity.

The main objective of the work is to evaluate if the proposed nonparametric approach using near surface air temperature as an additional predictor variable could improve radar precipitation estimates. This study compares the precipitation estimates using nonparametric model with temperature as a covariate to the model without temperature (univariate) and compares to the original precipitation rates using a constant Z-R relationship or separate rain (Z-R) or snow (Z-S) relationships. Further, we investigate if the improvements to the precipitation estimates varies with temperature ranges and if the method is dependent on the precipitation intensities.

*2) Some of the material in the Background is repeated from the Introduction. For example, the three paragraphs in section 2.1 (Radar precipitation estimation in cold climates) mostly in the Introduction. Either reduce the Introduction or merge the Background with the Introduction to remove the repetition. I suggest the latter and to consider adding sections to the Introduction (e.g., 1.1. Weather radar use for hydrology, 1.2. Radar precipitation estimation in cold climates, 1.3. Nonparametric Radar rainfall estimates).*

AC / CM: The Introduction and Background sections are integrated into a shorter and more specific Introduction to the paper as suggested by the reviewer.

*3) There are methods presented in the Study Area and Data section. These two sections 3. Methods and 4. Study Area and Data need to be revisited to put all the methods together. I suggest a brief section first on Study Area, then a section on Data and Methods, describing the data first, then the methods used.*

AC / CM: As the reviewer suggested, we put all the methods together under the Methodology section.

*4) The Results and Discussion are combined, and the Discussion is thus limited. I recommend that the Results and Discussion sections should be presented separately, or that the Discussion be much more in depth. There are only three citations in the entire Results and Discussion section, while numerous useful citations are presented in the Introduction and Background sections. There is no Discussion that put this work into context; the Results and Discussion only presents how do these results compare to the findings of Fassnacht et al. (1999), Koistinen et al. (2004) and Saltikoff et al. (2015).*

AC / CM: We have expanded the discussion as the reviewer suggested. It should, however, be noted that there are little similar works of this kind available and hence there are limited discussions that can be made to compare our results with those of earlier studies.

*5) At the end of the Summary and Conclusions, it is stated that "while this study uses data for one weather radar in arriving at its conclusions, preliminary analysis suggests the problems noted here to be generic." If there are additional "preliminary" results from some should be presented. This statement is important but is hanging.*

AC / CM: This was based on initial evaluation using X-band radar data, but since these data are not yet finally processed by the Norwegian Meteorological Institute, we decided to remove this statement.

*6) In various locations throughout the text, sentences are added that foreshadow the next or subsequent sections. These are not necessary and should be removed. The meshing of meteorological data with the radar data on a 1 x 1 km grid, including the interpolation of the station data is confusing. This is the four paragraphs on page 8 line 23 through page 9 line 21. This section needs to be rewritten, as it is unclear what is done. Perhaps a table could be added that describes the four datasets (T, RH, wind, and radar). From the text, I assume that temperature and RH data have been gridded at a 1-km resolution from the station data: T using the Optimal Interpolation in a Bayesian setting (Lussana et al., 2016) and RH using the nearest neighbor. What is the Lussana et al. (2016) method? While wind data are available, they are downscaled from a 10 km numerical model dataset. What numerical model dataset is used? This section needs to provide more details - it does not have to be much longer, the methods just need to be clarified. Also, the gauge precipitation (Pgauge) data are used as point measurements meshed with the gridded data; this would also be included in the aforementioned table. These data are revisited in section 4.1.*

AC / CM: We have removed the foreshadowing sentences where appropriate.

Regarding the other comments, the work presented in this paper uses 68-gauge locations and not grid locations. Since all gauges do not have air temperature, wind speed and relative humidity measurements, we used hourly gridded (1km x 1km) datasets from the Norwegian Meteorological Institute to generate air temperature and wind speed time series at the gauge locations. However, we do not have access to hourly gridded relative humidity (RH) data for the study area, but we do have measurements from 25 gauge locations. We used these gauge measurements where they were available at the location of the precipitation gauge, and for those gauges with missing RH we used measured RH from the closest gauge.

The hourly gridded (1 km x 1 km) air temperature and wind speed datasets were generated at the Norwegian Meteorological Institute. Lussana et al. (2016) spatially interpolated the past observed air temperature records from meteorological stations to develop the hourly gridded

temperature dataset for Norway. They used the Optimal Interpolation method in a Bayesian setting (Lussana et al., 2016). Norwegian Meteorological Institute derived the hourly gridded (1 km x 1 km) wind speed dataset by statistical downscaling from the 10 km numerical dataset, "NORA10" (documented in Reistad et al. (2011)) and "AROME" 2.5 km (documented in Müller et al. (2017)) using a local quantile regression (Lussana 2018, personal communication, 18 July).

As reviewer suggested, the following table (Table I) is added to describe the datasets used in the study and we rewrite the text to make this section clearer.

Table I: Different dataset used in this study and their source and spatial distribution

| Description | Gauge Precipitation | Radar precipitation | Air Temperature | Wind Speed | Relative Humidity |
|---|---|---|---|---|---|
| Spatial Distribution (Gridded / Gauge Locations) | At gauge locations | Gridded 1kmx1km | Gridded 1kmx1km | Gridded 1kmx1km | At gauge locations |
| Data Source | Gauge measurements | Radar measurements | Gauge measurements (spatially interpolated) | Downscaled from the numerical model ("NORA10" and AROME) | Gauge measurements |

*The font size is too small in most figures and the text is often grey. This makes the figures difficult to read. This should be addressed throughout.*
AC / CM: Font size is increased in the figures.

**Specific Comments:**
*Page 1, line 1: I suggest saying "In colder climates ..."*
AC / CM: "cold" is replaced with "colder"

*p1, l1 and l2, p2 l1, etc.: be consistent with "form of precipitation" and "state of precipitation."*
*I suggest calling it "phase of precipitation" throughout the text.*
AC / CM: Text is updated and the term "phase of precipitation" is used throughout the manuscript.

*p1, l5: "estimate" or "adjust"?*
AC / CM: It should be "adjust" and text is corrected.

*p1, l11: usually "catch error" is called "undercatch"*
AC / CM: The term "undercatch" is used throughout the manuscript.

*p1, l15: do you mean gauge air "temperature" or temperature at the radar measurement height?*
AC / CM: In this study, we used the air temperature at the gauge level. Text is updated to state this.

*p1 l15 and subsequently: to be more specific, use "warmer" than instead of "above" when referring to air temperatures. "Above" implies an altitude above the ground, which is typically associated with a colder air temperature. p2, l25: use "colder than" instead of "below," etc.*
AC / CM: Text is reworded.

*p1, l15-16: the end of the sentence "which indicates that the partial dependence of precipitation on air temperature is most important for colder climates alone" is unclear. Please reword.*
AC / CM: Text is reworded as follows.
"which indicates that the partial dependence of precipitation on air temperature is most useful for air temperatures below 10° C"

*p1, l22: should "2010b" be "2010a?" Check this, as (Villarini and Krajewski, 2010a) has not appeared yet.*
AC / CM: It is corrected.

*p2, l13: why "Conventionally?" Use another word so that the reader does not confuse radar types, such as "the original way" (i.e., conventional), Doppler, dual-polar, multiwavelength.*
AC / CM: "Standard approach" can be a right word here. Text is reworded.

*p2, l18: since Canada is mentioned here (Crozier et al., 1991) could be add to the citation list on line 20*
AC / CM: The citation (Crozier et al., 1991) is added.

*p2, l25: add an "s" to "quarter"*
AC / CM: It is corrected.

*p3, l2: "different temperatures cause different shapes of crystals." For solid precipitation, i.e., snow, the degree of super-saturation also affects the crystal shape.*
AC / CM: We acknowledge that degree of supersaturation also affects the crystal shape. Text is updated.

*p3, l5: what is meant by "multiple snow types?" Does this imply shapes? If so, state this explicitly.*
AC / CM: Snow type refers not only the shape of the snowflakes but also the particle density (Saltikoff et al., 2015). Text is updated.

*p3, l8: in many cases the correlation between probability of snow and temperature is an "'S' shaped structure," (see Fassnacht et al., 2001 for a summary illustration), but a simpler linear relation has also been used (e.g., Fassnacht et al., 2013).*
AC / CM: We agree with the reviewer that a simple linear relation has also been used but we mentioned the general pattern. Text is updated to include linear relation information.

p3, l9-10: be specific with "the dielectric property of solid particles (ice) is very different from liquid particles (water)." "Very different" is vague.
AC / CM: The text is reworded.

*p3, l12: reverse the order of the Hasan et al. (2016) references. You present 2016b before 2016a.*
AC / CM: It is corrected.

*p3, l13: change the word "Historical"*
AC / CM: Historical is changed with "past observed".

*p3, l25-29: delete the sentences in the rest of the paragraph starting with "the rest of the paper is structured as follows." You do not need to tell what the sections of the paper are, that reads like the end of a thesis. Instead, give us specific objectives to investigate or research questions that are answered.*
AC / CM: As per the reviewer's suggestion, we delete the sentence and the text is updated to list specific objectives of the work. Also see response above.

*p3, l28: The results should be presented, then there should be a separate Discussion section.*
AC: As mentioned above, there are little similar works of this kind available and we therefore find the integrated Results and Discussion to be useful.

*p4, l27: there has also been some work on phase discrimination using multiple radar wavelengths (e.g., Al-Sakka et al., 2013).*
AC/ CM: Thank you for pointing to a relevant work. We use and cite this paper in the revised manuscript.

*p4 l29 to p5 l3: This paragraph can be reduced to 1-2 sentences, as this information is generally known.*
AC / CM: We update the paragraph to make it more succinct.

*p5, l12-13: please reconsider "nonparametric approaches ... weakness is that the method is sensitive to outliers." I am not sure that this is correct. Parametric approaches tend to be sensitive to outliers.*
AC: Nonparametric approaches result in "local" biases as a result of outliers but the effect on global attributes is limited. On the other hand, parametric alternatives can be impacted globally due to biases in the estimated parameters.
CM: We reword the sentence in the revised manuscript.

*p5, l18-20: these two foreshadowing equations are not necessary.*
AC / CM: The two foreshadowing sentences are removed.

*p7, l4: "classification of precipitation phase at gauge level" is good, but don't we need the phase of precipitation at the radar height to select the appropriate radar Z-R equation? Although this is what Koistinen et al. (2004) and Saltikoff et al. (2015), it doesn't necessarily make it correct.*
AC: We agree with the reviewer that phase of precipitation at radar measurement height can be different from estimated phase at gauge height. We responded above in detail.

*Figure 1: a) I assume that the "length of the observations" is the number of hours with precipitation? b) I also assume that the hypsometry curve is cumulative % of stations below the specified elevation. Please be specific. c) the font size is small and difficult to read. Enlarge and also don't use grey. d) are the red names local cities? Are they important? If so, move them so they are legible.*
AC / CM: a) Yes, it is the number of hours with precipitation b) Yes, elevation of gauge locations, c) Font size is increased to make them readable, d) Yes, the cities' names are not important, and the figure and text are updated.

*p7, l12-13: What is the "accumulated hourly radar precipitation rate product?" Is this accumulated from sub-hourly to yield an hourly total, or is the hourly data added?*
AC / CM: This is accumulated from sub-hourly to yield an hourly total. Refer p8, l20.

*p7, l14: tell us how many gauges in the "a relatively dense network of precipitation gauges."*
AC / CM: We change the text as follows "a network of 88 precipitation gauges"

*p8, l26: instead of "are" use past tense through the methods.*
AC / CM: It is corrected.

*p8, l34: reword the last sentence "However, we used data from all available precipitation gauges for this study." Perhaps state something about the total number of gauge hours of data used (likely in the order of 100,000 gauge-hours).*
AC: Nearly 103000 total gauge hours were used in this study.
CM: Text is updated with total gauge hours.

*p9, l2: provide a source for the "gridded hourly temperature and wind speed dataset"*
AC: The source for gridded hourly temperature is observations from the Norwegian meteorological stations (refer p9, l7-8.)
The source for gridded wind speed data is the "NORA10" (documented in Reistad et al. (2011)) and "AROME" 2.5 km (documented in Müller et al. (2017)).
CM: Text is updated with source for gridded wind speed data. See response to comment 6) above for more detail.

*p9, l6: delete "more details on the procedure adopted for catch correction are provided in the next sub-section."*
AC/ CM: This sentence is deleted.

*p9, l10: change "resulted" to "resulting"*
AC / CM: "resulted" is replaced with "resulting".

*p9, l29: how little is "intensities below 0.1 mmh-1 contributes little?"*
AC: The percentage (quantity) is not mentioned in the cited work (Engeland et al., 2014). However, analysis of the data used in this study showed that intensities lower than 0.1 mmh$^{-1}$ and greater than 0.05 mmh$^{-1}$ are nearly 10 % of the total data above 0.05 mmh$^{-1}$.

*p10, l2-8: this is background. It could be moved to earlier in the text, as this is the methods/data section. Tell us what was done. This sentence could also be deleted.*
AC / CM: As reviewer suggested, we move it to earlier in the text and update.

*p10, l4: the word "Nordic" is not necessary here, as it could also be relevant in southern environments*
AC / CM: We agree. "Nordic" is deleted in the revised manuscript.

*p10, l4-5: the end of the sentence is redundant "due to large catch errors for snow." In could state that it is "due to high wind conditions." It is wind that causes undercatch, not "large catch errors"*
AC / CM: Text is reworded.

*p10, l7: "Wolff et al., 2015" is not in the citation list*
AC / CM: References is updated with "Wolff et al. (2015)"

*p10, l9 or previous: what type of precipitation gauge and what type of shield are used? This is very important information to assess the degree of undercatch and the error associated with the undercatch correction.*
AC / CM: "The gauges in the study site consists of "Geonor" type and tipping bucket. Both types are with an Alter wind shield. This information is added to the manuscript.

*p10, l9-17: throughout this paragraph it is "undercatch" correction. This should be consistent, as there are other errors.*
AC / CM: We agree. Text is updated.

*p10, l13-14: "It was found that correlation between the corrected precipitation by using measured wind speed data (15-gauge locations) and gridded data are over 0.97..." Does this mean the correlation undercatch correction using gauge wind speed versus the downscaled gridded wind speed?*
AC: Yes, the correlation between undercatch corrected precipitation using gauge wind speed versus undercatch corrected precipitation using the downscaled gridded wind speed. This was done to verify that gridded wind would provide a realistic correction compared to wind measured at the site.

*p10, l14: there are only 15-gauge locations with wind speed measurements. How are the other 53 precipitation gauges corrected for undercatch? From my comment above (p10, l13-14), I assume that the downscaled gridded wind speed was used to correct for undercatch at all 68 precipitation gauges. This is not clear.*

AC: The downscaled gridded wind speed was used to correct for undercatch at all 68 precipitation gauges. To control the result of correcting with gridded wind speed, we compared the corrected precipitation using gridded wind speed with the 15 locations where we had wind speed measurements at the gauge site.

*p10, l19-22: delete this paragraph. We know what you are going to do Figure 2: add tick marks to the y-axis.*

AC / CM: The foreshadowing paragraph is deleted. Tick marks are added to the y-axis.

*p10, l23 through p11: Are the "Partial weight of predictors" constant over time, i.e., is there a specific value for station that does not change?*

AC: No, we did not use a fixed value for any given station. The value varies with the data.

*Table 1: As this is a summary of Figure 2, this table could be converted to two horizontals box and whisker plots on Figure 1. The reader doesn't really care about the specific numbers, just the range.*

AC: As the reviewer suggested we have plotted box and whisker plot (refer Fig. I in Appendix). However, we feel Table is a better representation of the information we wish to convey and we prefer to keep the Table 1 in the revised manuscript.

*p12, l1 and elsewhere: the word "prediction" implies that this is for the future. I suggest using "estimation" throughout.*

AC / CM: "Prediction is replaced with "estimation".

*Figure 3: the caption is confusing; break into two sentences. Also, are these "length of the data (circle size)" the same as in Figure 1? If so, then don't display again here, unless this is relevant later?*

AC / CM: The caption is reworded.
Yes, Length of the data (circle size) is same on both Fig. 1 and Fig. 3.
Fig.3 is updated.

*p13, l4-7: these three sentences could be reduced to a small histogram of RMSE reductions that is added to figure 3.*

AC: As the reviewer suggested, we have plotted a histogram (refer Fig. II in Appendix). However, we prefer to keep these sentences for simplicity in the revised manuscript as well.

*p12 to 15: why was a single average (mean) partial weight (beta P = 0.68) used in the k-nn prediction model, when it was shown (Figure 2 and Table 1) that the partial weight varies from 0.4 to 1?*

AC: We responded above in detail.
However, the use of station specific partial weight can be more precise when sufficient past observed data are available at each gauge location.

*p15 section 5.3: all this text except for the last sentence is background or methods and should be moved to an appropriate location earlier in the paper.*

AC: We think that the first paragraph of section 5.3 present information on the data that is relevant for the understanding of the low intensity analysis and figure 5. We would therefore rather keep this section as it is in the paper.

*p15, l15: does this "still significant" has a statistical meaning? If so, explain how. If not don't use the word significant.*

AC / CM: Yes, the results are statistically significant as the RMSE was estimated using leave one out cross validation (LOOCV). We clarify this in the revised manuscript.

*p15, section 5.4: what is the range of the "Temperature Classes?" You only discuss T > 10C. What about T < 10C? The point of this section is unclear.*

AC / CM: The results for T < 10° C is added and paragraph is updated.

*Figure 6: what are the dots above the solid and mixed phase?*

AC / CM: The dots symbolise outliers, values outside 1.5 * IQR which is represented by the whiskers. An explanation is added to the figure caption.

*p18, section 5.6: these statements seem to be hanging. Can you present specifics?*

AC / CM: We did a test on uncorrected gauge precipitation data (not corrected for wind induced undercatch) during an early phase of the study and found that air temperature works as a covariate also there. For this assessment, we used the uncorrected gauge precipitation at 88 gauge locations. More than 80 % of the precipitation gauge locations in the study area showed clear improvement. The intent of section 5.6 was to make this point clearly. However, in order to avoid lengthening the paper with more results, we decided to remove section 5.6 in the revised manuscript and merge the statements into other sections.

*p18, l5: what does "the use of temperature as an additional predictor variable is having consistent impact" mean? The words "consistent impact" are not clear.*

AC: We mean by "consistent impact" that the study with uncorrected gauge precipitation as described above resulted in similar results (resulted in partial weight for air temperature and improvement in RMSE with air temperature as an additional predictor) as like the study with corrected gauge precipitation.

**References**

Al-Sakka, H., Boumahmoud, A.-A., Fradon, B., Frasier, S. J. & Tabary, P. 2013. A New Fuzzy Logic Hydrometeor Classification Scheme Applied to the French X-, C-, and S-Band Polarimetric Radars. *Journal of Applied Meteorology and Climatology,* 52**,** 2328-2344.

Crozier, C., Joe, P., Scott, J., Herscovitch, H. & Nichols, T. 1991. The king city operational doppler radar: Development, all-season applications and forecasting. *Atmosphere-Ocean,* 29**,** 479-516.

Fassnacht, S., Soulis, E. & Kouwen, N. 1999. Algorithm application to improve weather radar snowfall estimates for winter hydrologic modelling. *Hydrological processes,* 13**,** 3017-3039.

Fassnacht, S. R., Kouwen, N. & Soulis, E. D. 2001. Surface temperature adjustments to improve weather radar representation of multi-temporal winter precipitation accumulations. *Journal of Hydrology,* 253**,** 148-168.

Koistinen, J., Michelson, D. B., Hohti, H. & Peura, M. 2004. Operational Measurement of Precipitation in Cold Climates. *In:* MEISCHNER, P. (ed.) *Weather Radar: Principles and Advanced Applications.* Berlin, Heidelberg: Springer Berlin Heidelberg.

Lussana, C., Ole, E. T. & Francesco, U. 2016. seNorge v2.0: an observational gridded dataset of temperature for Norway. Norway: Norwegian Meteorological Institute.

Müller, M., Homleid, M., Ivarsson, K.-I., Køltzow, M. a. Ø., Lindskog, M., Midtbø, K. H., Andrae, U., Aspelien, T., Berggren, L., Bjørge, D., Dahlgren, P., Kristiansen, J., Randriamampianina, R., Ridal, M. & Vignes, O. 2017. AROME-MetCoOp: A Nordic Convective-Scale Operational Weather Prediction Model. *Weather and Forecasting,* 32**,** 609-627.

Reistad, M., Breivik, Ø., Haakenstad, H., Aarnes, O. J., Furevik, B. R. & Bidlot, J.-R. 2011. A high-resolution hindcast of wind and waves for the North Sea, the Norwegian Sea, and the Barents Sea. *Journal of Geophysical Research: Oceans,* 116.

Saltikoff, E., Lopez, P., Taskinen, A. & Pulkkinen, S. 2015. Comparison of quantitative snowfall estimates from weather radar, rain gauges and a numerical weather prediction model. *Boreal Env. Res,* 20**,** 667-678.

Tveito, O., Førland, E., Heino, R., Hanssen-Bauer, I., Alexandersson, H., Dahlström, B., Drebs, A., Kern-Hansen, C., Jónsson, T. & Vaarby Laursen, E. 2000. Nordic temperature maps. *DNMI report,* 9.

Tveito, O. E. & Førland, E. J. 1999. Mapping temperatures in Norway applying terrain information, geostatistics and GIS. *Norsk Geografisk Tidsskrift-Norwegian Journal of Geography,* 53**,** 202-212.

Wolff, M. A., Isaksen, K., Petersen-Øverleir, A., Ødemark, K., Reitan, T. & Brækkan, R. 2015. Derivation of a new continuous adjustment function for correcting wind-induced loss of solid precipitation: results of a Norwegian field study. *Hydrol. Earth Syst. Sci.,* 19**,** 951-967.

**Appendix**

[Figure]

**Figure I.** Box and whisker plot of estimated partial weight of predictor variables (Radar precipitation rate and air temperature) at 68 gauge locations in the study area. The summation of partial weights is equal to 1.

[Figure]

**Figure II.** Percentage of precipitation gauge locations against percentage improvement in RMSE with air temperature as an additional predictor variable at those gauge locations and the mean RMSE improvement percentage (red dash line) for gauge locations (68 gauges) in the study area.

---

## Author Comment (AC2) · 27 Aug 2018

**Response to the review of hess-2018-0351**

**RC2: Responses to Anonymous Referee 2**
The authors wish to thank the reviewer for his constructive comments and corrections to the discussion paper. In the following, we have responded to each of the comments from the reviewer for this manuscript. The comment from the reviewer (RC) is in italic font while the author comment (AC) and changes in the manuscript (CM) are in blue normal font.

*General comments:*
*The authors apply the non-parametric k - nearest neighbour method (k-nn) to estimate radar precipitation from gridded surface observations of rainfall and temperature for the Oslo region in Norway. They show that utilising temperature as second predictor variable reduces the root mean squared error significantly compared to a k-nn model without temperature and compared to the original procedure using a constant Z-R relationship or separate snow/rain Z-R relationships.*
*The application of this method for radar rainfall estimation including temperature is novel and of interest not only for readers living in regions with colder climates. The research is done systematically and quite carefully. The paper is written well and clear in structure. However, there are three major points and some minor things which need attention before the paper can be published. One main point are the lengthy introduction and background sections which could be shortened. A second important point concerns the method to estimate the partial weights. It becomes not clear, that this method is really providing optimal weights. And, third, there seems to be an issue with the back-calculation of Z using the inverse Z-R relationship on a different time resolution as for the original forward calculation. Detailed information about this and the minor things are given below.*

*Detailed comments:*
*1. Sections 1 and 2: Both sections together cover almost 4 pages and represent the introduction with the state of the art. This is quite lengthy. The introduction is very general; the background is more focussed on the topic at hand. I would suggest to shorten these parts especially the introduction significantly and may be use the background as introduction.*
AC / CM: We have merged the Introduction and background sections as suggested and updated the text to make the introduction more succinct.

*2. Eq. 1: As predictor R(t) is used. Why not using Z(t) as predictor? For R(t) already a (wrong) Z-R-relationship has been applied, introducing great uncertainty. If a linear relationship is required a log-log transformation of Z(t) and Rest(t) could be applied beforehand. This needs at least to be discussed.*
AC / CM: In the methodology presented in the paper, reflectivity (dBZ) could in principle be used instead of radar precipitation rate as shown by Hasan et al. (2016) for the univariate case. As we do not have access to the reflectivity (Z(t)) data from the Hurum radar for this study, we had to use the hourly radar precipitation rate which is available from the Norwegian Meteorological Institute. While the reflectivity can be back-calculated by inverting the algorithm that was used operationally, we feel this may add additional uncertainty and would not matter given the regression algorithm being used is nonparametric. Further, it can be noted that one key purpose of this work is to see how we can improve the radar precipitation rate data available to us as a finished product (hourly Surface Rainfall Intensity (SRI) product) from the meteorological institute.
The text is updated in the revised manuscript to clarify this.

*3. Fig. 1: The units for observation length and elevation are missing. Also, the text of the legend is tiny and hard to read.*

AC / CM: The units are added, and the font size of the text is increased in Fig.1.

*4. Section 5.1: It is not clear if the estimation of the partial weights using partial information correlation (PIC) is really beneficial or even optimal. In order to prove the merit of PIC I would suggest to test two additional cases a) equal weights for P and T and b) using simple linear partial correlations. The performance for the latter two cases measured by RMSE should be worse than by PIC weighing.*

AC / CM: The partial informational correlation (PIC) provides a generic measure of statistical dependence of predictors of a general linear or nonlinear system. Estimation of partial weight using PIC shows the partial dependence of radar precipitation estimation on air temperature. Earlier papers have shown that the estimated PIC and weights collapse to what would be estimated using a linear regression model if the system is linear (Mehrotra and Sharma, 2006, Sharma and Mehrotra, 2014, Sharma et al., 2016). As the system here is nonlinear, our approach of using PIC to estimate partial weights appears more justified.

After receiving the reviewer's comment, we tested our approach using equal weights. We found that the gain in RMSE was not significant with the use of PIC based partial weight compared to equal weights, but the mean error was reduced when we used partial weight estimated using PIC.

The manuscript is updated to discuss this.

*5. Fig. 4: This bar plot is not easy to read. I would suggest to use box-whisker plots instead.*

AC:  As the reviewer suggested we have plotted box and whiskers plot (refer Fig. I Appendix). However, bar plot presents the results at each gauge location compared to lumped box plot which we find interesting to report so we would like to retain the bar plot in the manuscript. We have attempted to improve the bar plot to make it more readable.

*6. Page 16, line 1: The back-calculation of Z from R using a non-linear relationship on hourly data gives an estimated average Z value for each hour. This estimate can be quite different from the observed average Z value if the rainfall distribution within the hour is not unique. In the forward calculation the Z-R relationship is applied on 7.5 min Z values to calculate 7.5 minute rainfall intensities. Because of the non-linearity of the Z-R relationship a simple back calculation on a different time step than the one the original calculation was applied is not possible. For non-linear functions f is $E[f(x)] <> f[E(x)]$.*

AC / CM: We do agree with the reviewer that back calculated reflectivity on a different time step (hour) than the original calculation (7.5 minutes) is not same as the average value unless the precipitation is even within the hour and we fully acknowledge that this introduces uncertainties in the results.  As mentioned above, we do not have access to reflectivity data (or 7.5 minutes precipitation rates). In order to compare our proposed nonparametric radar precipitation estimation with radar precipitation rates computed using separate equations for snow and rain, we decided to back calculate the reflectivity from the data available to us.

The manuscript is updated to discuss the uncertainty in the computation in more detail.

**References**

Hasan, M. M., Sharma, A., Johnson, F., Mariethoz, G. & Seed, A. 2016. Merging radar and in situ rainfall measurements: An assessment of different combination algorithms. *Water Resources Research,* 52**,** 8384-8398.

Mehrotra, R. & Sharma, A. 2006. Conditional resampling of hydrologic time series using multiple predictor variables: A K-nearest neighbour approach. *Advances in water resources,* 29**,** 987-999.

Sharma, A. & Mehrotra, R. 2014. An information theoretic alternative to model a natural system using observational information alone. *Water Resources Research,* 50**,** 650-660.

Sharma, A., Mehrotra, R., Li, J. & Jha, S. 2016. A programming tool for nonparametric system prediction using Partial Informational Correlation and Partial Weights. *Environmental Modelling & Software,* 83**,** 271-275.

**Appendix**

[Figure]

**Figure I:** Box plot representing three quality metrics (RMSE, MAE and ME) at gauge locations for the original data (MP) and for the two nonparametric models (knn-R and knn-RT). Mean value of quality metrics for each model by red diamond point. Here, knn-R denotes the nonparametric model with radar precipitation rate as a single predictor, while knn-RT denotes the nonparametric model with radar precipitation rate and air temperature as two predictors with fixed partial weight of (0.68, 0.32).

---

## Author Comment (AC3) · 27 Aug 2018

The comment was uploaded in the form of a supplement:
https://www.hydrol-earth-syst-sci-discuss.net/hess-2018-351/hess-2018-351-AC3-supplement.pdf

---

## Author Comment (AC4) · 27 Aug 2018

**Response to the review of hess-2018-0351**

**RC3: Responses to G. Ravazzani (Referee 3)**
The authors wish to thank the reviewer for his constructive comments and corrections to the discussion paper. In the following, we have responded to each of the comments from the reviewer for this manuscript. The comment from the reviewer (RC) is in italic font while the author comment (AC) and changes in the manuscript (CM) are in blue normal font.

*General comments:*
*In this paper, a non-parametric method is applied to estimate radar precipitation considering both rainfall and temperature. The use of radar for precipitation estimation is an interesting topic. Many papers have been presented about this topic, but the specific problem authors deal in this paper is how to assess solid precipitation in cold regions. The solution they propose is of interest for cold climates in northern Europe, of course, but I suppose it could be extended to other areas where solid precipitation occurs.*

*Specific comments:*
*Authors used 68 rain gauges in this study that are clustered around urban areas. Do authors think that this uneven distribution may affect results? In other terms, is the location of raingauges relevant for the application of the proposed procedure?*
AC: The method used is independent of the gauge locations, and the computed estimates are ascertained for each gauge individually.

*P 9 L 14 "The gridded hourly wind speed datasets are derived from a statistical downscaling of a 10 km numerical model dataset onto a 1 km grid". Did authors verify how the method is sensible to the specific realization of the statistical downscaling?*
AC: The Norwegian Meteorological Institute derived the hourly gridded (1 km x 1 km) wind speed dataset by statistical downscaling from the 10 km numerical dataset, "NORA10" and we used this in this study. We did not evaluate their method of downscaling. However, as described on p10, l12-14, to control the result of correcting with gridded wind speed, we compared the corrected precipitation using gridded wind speed with the 15 locations where we have wind speed measurements at the gauge site.

*Authors apply correction to gauge precipitation to consider wind induced underestimation. Gauge precipitation is affected by several sources of uncertainty. Wind is of course relevant, but another systematic error is related to the calibration of raingauges that causes underestimation for high rainfall intensity and overestimation for low rainfall intensity. Further uncertainty arises when solid precipitation has to be measured. How did authors deal with these errors? Are they already managed by the meteorological institute?*
AC: Norwegian Meteorological Institute manages the calibration of raingauges and takes necessary measures to reduce the uncertainty that arises when solid precipitation has to be measured. Further, data from the raingauges are gone through routine quality control before being released to the public through the database. However, the meteorological institute does not do wind induced undercatch correction for their precipitation data.

*Section 5.6 is very short compared to the rest of the paper and I did not fully understand what is the intention of authors. I think they should better explain this part or remove it.*
AC / CM: we did a test on uncorrected gauge precipitation data (not corrected for wind induced undercatch) showing that temperature works as a covariate also there. The intent of section 5.6 was to make this point clearly. However, in order to avoid lengthening the paper with more

results, we decided to remove section 5.6 in the revised manuscript and merge the above statement into other sections.

*Technical corrections:*
*P.4 L. 6 The Finnish Meteorological Institute*
AC / CM: It is corrected.

---

## Author Response (AR1)

**Dear Editor,**

Thank you for your constructive comments to the discussion paper and for the opportunity to submit the revised paper. We have answered all the comments from reviewers in the authors' responses and revised the manuscript accordingly. The revised paper is significantly improved as a result of addressing these comments as we now have assessed the sensitivity to our findings to all factors that could possibly be impacting the results.

As the reviewers and the editor suggested, we have restructured the manuscript into the standard IMRAD format. We have merged the introduction and background sections as suggested and updated the text to make the introduction shorter and more succinct. After reading the review, we think that the description of data (including the source and methods to generate the data) used in the study was not clear enough. It can be noted that the gridded temperature and wind speed were generated for all of Norway by the Norwegian Meteorological Institute and we used this data in our study. In this revision, we tried to better describe all the data sources used within the subsection "Data" under the section "Materials and Methods".

The methodology and the context of the study as well as main results remain unchanged. We have added a separate discussion section to the paper. As the reviewers and editor mentioned, we agree that the justification for the approaches used in the study was limited in the discussion paper even though we feel we described the methodology clearly. In the revision, we have used the Discussion section to provide proper justification for the approaches used and to discuss the uncertainties and limitations in the study.

Reviewer 2 pointed out that the back calculation of reflectivities from the hourly radar precipitation data originally based on accumulated data of 7.5 minutes would only be correct if the precipitation is even within the hour. Unfortunately, the reflectivity data used to produce the radar precipitation rates (SRI product) used in the study are not stored in the production process, and therefore not available at the Norwegian Meteorological Institute (met.no) (Elo 2018, Personal communication). However, Plan Position Indicator (PPI) of the lowest elevation beam from Hurum radar with the original short time resolution is available from met.no and in the revised paper we have repeated our computations for the comparison of the proposed nonparametric radar precipitation estimation with radar precipitation rates computed using separate equations for snow and rain.

In the revised manuscript, we have used the PPI data to redistribute the hourly data (SRI) by assuming precipitation intensity distribution within each hour is as same for SRI as PPI. The redistributed precipitation rates were then converted to reflectivities and these data are used for the analysis as in the original manuscript. It should be noted that there is uncertainty in how accurately the redistributed intensity distribution of SRI represents the original distribution, however, this exercise at least used a possible realistic distribution.

We believe that we have revised the manuscript as described in the authors' responses to reviewers and in the case of using different Z-R relationship for snow and rain, the availability of the PPI data made it possible to improve the analysis beyond what is discussed in the response to the reviewer2.

According to HESS requirements, this submission consists of the revised manuscript and a point-by-point reply to the comments to the three reviewers and a marked-up manuscript version showing the changes made. In case, if you need any clarifications or further details,

please feel free to contact us. We hope that Hydrology and Earth System Science will find it an interesting contribution.

With thanks,

Sincerely, Kuganesan Sivasubramaniam Ashish Sharma Knut Alfredsen

**Response to the review of hess-2018-0351**

**RC1: Responses to S.R. Fassnacht (Referee 1)**

The authors wish to thank the reviewer for his constructive comments and corrections to the discussion paper. In the following, we have responded to each of the comments from the reviewer and showed the page and line numbers of the revised manuscript if any changes. The comment from the reviewer (RC) is in italic font while the author comment (AC) and changes in the manuscript (CM) are in blue normal font.

This is an interesting paper that should give us some improved insight into using weather radar to estimate rain and snow fall. This is very relevant in higher latitudes and in mountain environments were snow is important. To date, there has been limited work in using weather radar for snowfall estimation in a hydrological context. The methods presented herein could be used in many locations. However, the writing is unclear, and I got lost at times. I suggest that the authors revisit their objectives and ensure that the paper addresses these. Also, the Discussion is essentially missing as the work is not put into context of the few other relevant studies. Below I outline restructuring and a problem with the Methods/Data.

Equation 4 uses air temperature and relative humidity to estimate the phase of the precipitation from Koistinen et al. (2004) and used by Saltikoff et al. (2015) for Finland. However, air temperature at the gauge is used, and this is not correct. Fassnacht et al. (1999; 2001) lapsed the air temperature up to the radar measurement height. There can be 5 to 10 degrees Celsius difference between the temperature at the height (2 m above the ground) and the radar height (1 km stated on page 8 line 15). At minimum this should be discussed?

AC: As it is mentioned, we adopted the operational method from the Finnish Meteorological Institute as presented by Koistinen et al. (2004) and Saltikoff et al. (2015). In their phase equation, near surface temperature (2m above ground) is defined and we did follow their method as defined in the papers. We do agree with the reviewer that air temperature at radar measurement height can be different from gauge height and hence the estimated phase can be different. This has already been discussed on p17, I12-13 "Further, our phase classification is at gauge level, and represents near surface conditions. The phase of the precipitation can be different at the elevation where the radar measures the reflectivity."

Air temperature can be lapsed to the radar measurement height to estimate the phase of precipitation. Fassnacht et al. (1999) and Fassnacht et al. (2001) assumed the temperature lapse rate to be zero in their studies as winter lapse rate is often zero in mid latitude areas. For the Nordic region, Tveito and Førland (1999) showed that vertical lapse rate varies with season and location. Further, Tveito et al. (2000) found that local terrain conditions have greater influence in local temperature gradient during winter. Due to the occurrence of inversions, lapse rate can deviate substantially from the standard (-6.5° C/km) during the winter months and it can be as low as -1.2° C/km (Tveito et al., 2000, Tveito and Førland, 1999). The estimated temperature at radar measurement height and hence the probability of liquid phase (Plp) are therefore highly uncertain. The measurements of phase information at radar measurement height with the use of dual polarized radars can be a useful data source for further investigation.

After receiving the reviewer's comment, we investigated the sensitivity of our results to the use of a lapse rate. The air temperature at the radar measurement height (1 km) was computed using a standard moist adiabatic lapse rate (-6.5° C/km) as used in Nordic meteorological studies (Tveito et al., 2000). We estimated the probability of liquid precipitation (PIp) in Eq. (4) by using the lapsed temperature at radar measurement height while assuming the relative humidity unchanged. The estimated PIp was used to classify the precipitation phase and repeated the work as presented in section 5.5. Our results (not included in the revised paper) showed that new classification did not improve RMSE compared to the use of near surface

phase classification. We attribute this to the considerable uncertainty associated with the use of the lapse rate as noted by others (Al-Sakka et al., 2013, Tveito et al., 2000). Further, equation (Eq. (4)) is developed and tested for surface phase classification and it incorporates relative humidity as a variable. We assumed relative humidity at the radar measurement height similar to the surface value. This is also a potential error source in the computation.

We, therefore want to keep the Finnish Meteorological Institute's operational method of surface phase estimation to classify the precipitation as the method of choice in the paper. This is both in operational use and developed for the Nordic area which gives us some confidence in the method. We discuss this issue in the revised manuscript. CM: p18, I11-20

What about a split sample approach of calibration and evaluation for the partial weights and *k*-nn approach? Also, the partial weight for radar precipitation was shown to vary from 0.4 to 1 (Figure 2), so why was a single average (mean) used in the *k*-nn prediction model. Is this approach not robust enough to have a different partial weight, or perhaps a gridded partial weight? It is stated that there is no spatial pattern in the partial weights, but an interpolated residual type approach could be used (e.g., Fassnacht et al., 2003 among others).

AC: As described on p12, I15-17, a split sample test was done to verify the results obtained from the leave one out cross validation (LOOCV) approach and presented in the paper.

Partial weights did not show any spatial pattern that would allow us to generate an informed specification of the weights that could be applied over the study region. Further, the RMSE estimated at gauge locations with the single average partial weight for the study area (as we presented in this paper) showed a strong resemblance with the RMSE estimated by using the partial weight estimated from the five nearest gauges. Hence, we decided to use a single average partial weight to present in this paper.

As the reviewer mentioned, it is possible to use gridded partial weight or an interpolated residual type approach. However, we found that the gain in RMSE is not significant for the effort of using gridded or residual type partial weights. The added complexity of gridding the partial weights does not add significant information to the analysis and we therefore recommend using an average value in the computations for this study region. CM: p17, I17-19

The paper does need restructuring and rewriting. At present I get lost in where I am in the text, regardless of the "foreshadowing" sentences that appear at the end of various sections. 1) At the end of the Introduction, the paper should tell the reader specific objectives that were investigated or research questions that were answered.

AC: As per the reviewer's suggestions we have added specific objectives at the end of the introduction.

CM: p4, l1-8

2) Some of the material in the Background is repeated from the Introduction. For example, the three paragraphs in section 2.1 (Radar precipitation estimation in cold climates) mostly in the Introduction. Either reduce the Introduction or merge the Background with the Introduction to remove the repetition. I suggest the latter and to consider adding sections to the Introduction (e.g., 1.1. Weather radar use for hydrology, 1.2. Radar precipitation estimation in cold climates, 1.3. Nonparametric Radar rainfall estimates).

AC: The Introduction and Background sections are integrated into a shorter and more specific Introduction to the paper as suggested by the reviewer. CM: p1-p4, section 1

3) There are methods presented in the Study Area and Data section. These two sections 3. Methods and 4. Study Area and Data need to be revisited to put all the methods together. I suggest a brief section first on Study Area, then a section on Data and Methods, describing the data first, then the methods used.

AC: As reviewer suggested, we have added a section "Materials and Methods" where a brief subsection first on Study area, and then Data followed by Methodology. CM: From p4, I9 to p9, I15

4) The Results and Discussion are combined, and the Discussion is thus limited. I recommend that the Results and Discussion sections should be presented separately, or that the Discussion be much more in depth. There are only three citations in the entire Results and Discussion section, while numerous useful citations are presented in the Introduction and Background sections. There is no Discussion that put this work into context; the Results and Discussion only presents how do these results compare to the findings of Fassnacht et al. (1999), Koistinen et al. (2004) and Saltikoff et al. (2015).

AC: As reviewer suggested, we have added a separate Discussion section. In this section, we provided proper justification for the approaches used and discussed the uncertainties and limitations in the study.

CM: From p16, I1 to p18, I34

5) At the end of the Summary and Conclusions, it is stated that "while this study uses data for one weather radar in arriving at its conclusions, preliminary analysis suggests the problems noted here to be generic." If there are additional "preliminary" results from some should be presented. This statement is important but is hanging.

AC / CM: This was based on initial evaluation using X-band radar data, but since these data are not yet finally processed by the Norwegian Meteorological Institute, we decided to remove this statement.

6) In various locations throughout the text, sentences are added that foreshadow the next or subsequent sections. These are not necessary and should be removed. The meshing of meteorological data with the radar data on a 1 x 1 km grid, including the interpolation of the station data is confusing. This is the four paragraphs on page 8 line 23 through page 9 line 21. This section needs to be rewritten, as it is unclear what is done. Perhaps a table could be added that describes the four datasets (T, RH, wind, and radar). From the text, I assume that temperature and RH data have been gridded at a 1-km resolution from the station data: T using the Optimal Interpolation in a Bayesian setting (Lussana et al., 2016) and RH using the nearest neighbor. What is the Lussana et al. (2016) method? While wind data are available, they are downscaled from a 10 km numerical model dataset. What numerical model dataset is used? This section needs to provide more details - it does not have to be much longer, the methods just need to be clarified. Also, the gauge precipitation (Pgauge) data are used as point measurements meshed with the gridded data; this would also be included in the aforementioned table. These data are revisited in section 4.1.

AC: We have removed the foreshadowing sentences where appropriate.

Regarding the other comments, the work presented in this paper uses 68-gauge locations and not grid locations. Since all gauges do not have air temperature, wind speed and relative humidity measurements, we used hourly gridded (1km x 1km) datasets from the Norwegian Meteorological Institute to generate air temperature and wind speed time series at the gauge locations. However, we do not have access to hourly gridded relative humidity (RH) data for the study area, but we do have measurements from 25 gauge locations. We used these gauge measurements where they were available at the location of the precipitation gauge, and for those gauges with missing RH we used measured RH from the closest gauge.

The hourly gridded (1 km x 1 km) air temperature and wind speed datasets were generated at the Norwegian Meteorological Institute. Lussana et al. (2016) spatially interpolated the past observed air temperature records from meteorological stations to develop the hourly gridded temperature dataset for Norway. They used the Optimal Interpolation method in a Bayesian setting (Lussana et al., 2016). Norwegian Meteorological Institute derived the hourly gridded (1 km x 1 km) wind speed dataset by statistical downscaling from the 10 km numerical dataset,

"NORA10" (documented in Reistad et al. (2011)) and "AROME" 2.5 km (documented in Müller et al. (2017)) using a local quantile regression (Lussana 2018, personal communication, 18 July).

As reviewer suggested, the following table (Table I) is added to describe the datasets used in the study and we rewrite the text to make this section clearer.

| Description                                                  | Gauge
Precipitation | Radar
precipitation | Air
Temperature                                   | Wind Speed                                                        | Relative
Humidity  |
|--------------------------------------------------------------|------------------------|------------------------|------------------------------------------------------|-------------------------------------------------------------------|-----------------------|
| Spatial
Distribution
(Gridded /
Gauge
Locations) | At gauge
locations  | Gridded
1kmx1km     | Gridded
1kmx1km                                   | Gridded 1kmx1km                                                   | At gauge
locations |
| Data
Source                                               | Gauge
measurements  | Radar
measurements  | Gauge
measurements
(spatially
interpolated) | Downscaled from
the numerical
model ("NORA10"
and AROME) | Gauge
measurements |

Table I: Different dataset used in this study and their source and spatial distribution

**CM: From p5, I3 to p7, I28, section 2.2**

The font size is too small in most figures and the text is often grey. This makes the figures difficult to read. This should be addressed throughout. AC / CM: Font size is increased in the figures.

**Specific Comments:**

Page 1, line 1: I suggest saying "In colder climates ..." AC / CM: The sentence has been deleted in the revised manuscript.

p1, I1 and I2, p2 I1, etc.: be consistent with "form of precipitation" and "state of precipitation." I suggest calling it "phase of precipitation" throughout the text. AC / CM: Text is updated and the term "phase of precipitation" is used throughout the manuscript.

p1, I5: "estimate" or "adjust"?

AC / CM: The sentence has been deleted in the revised manuscript.

*p1, I11: usually "catch error" is called "undercatch"* AC / CM: The term "undercatch" is used throughout the manuscript.

*p1, l15: do you mean gauge air "temperature" or temperature at the radar measurement height?*

AC: In this study, we used the air temperature at the gauge level. Text is updated to state this. CM: p1, I16

p1 I15 and subsequently: to be more specific, use "warmer" than instead of "above" when referring to air temperatures. "Above" implies an altitude above the ground, which is typically associated with a colder air temperature. p2, I25: use "colder than" instead of "below," etc. AC: Text is reworded.

CM: p1, l16

p1, I15-16: the end of the sentence "which indicates that the partial dependence of precipitation on air temperature is most important for colder climates alone" is unclear. Please reword. AC: Text is reworded. CM: p1, I15-17

p1, I22: should "2010b" be "2010a?" Check this, as (Villarini and Krajewski, 2010a) has not appeared yet. AC: It is corrected. CM: p1, I22

*p2, 113:* why "Conventionally?" Use another word so that the reader does not confuse radar types, such as "the original way" (i.e., conventional), Doppler, dual-polar, multiwavelength. AC: "Standard approach" can be a right word here. Text is reworded. CM: p2, I9

p2, I18: since Canada is mentioned here (Crozier et al., 1991) could be add to the citation list on line 20
AC: The citation (Crozier et al., 1991) is added.
CM: p2, I18

*p2, l25: add an "s" to "quarter"* AC: The sentence has been deleted in the revised manuscript.

p3, I2: "different temperatures cause different shapes of crystals." For solid precipitation, i.e., snow, the degree of super-saturation also affects the crystal shape.
AC: We acknowledge that degree of supersaturation also affects the crystal shape. Text is updated.
CM: p3, I4

CM: p3, l4

p3, I5: what is meant by "multiple snow types?" Does this imply shapes? If so, state this explicitly.

AC: Snow type refers not only the shape of the snowflakes but also the particle density (Saltikoff et al., 2015). Text is updated. CM: p3, I7-8

p3, l8: in many cases the correlation between probability of snow and temperature is an "S' shaped structure," (see Fassnacht et al., 2001 for a summary illustration), but a simpler linear relation has also been used (e.g., Fassnacht et al., 2013).

AC: We agree with the reviewer that a simple linear relation has also been used but we mentioned the general pattern. Text is updated to include linear relation information. CM: p3, I11

p3, I9-10: be specific with "the dielectric property of solid particles (ice) is very different from liquid particles (water)." "Very different" is vague. AC: The text is reworded. CM: p3, I12

p3, I12: reverse the order of the Hasan et al. (2016) references. You present 2016b before 2016a. AC: It is corrected. CM: p3, I29

p3, I13: change the word "Historical"AC: Historical is changed with "past observed".CM: p3, I31

p3, I25-29: delete the sentences in the rest of the paragraph starting with "the rest of the paper is structured as follows." You do not need to tell what the sections of the paper are, that reads like the end of a thesis. Instead, give us specific objectives to investigate or research questions that are answered.

AC: As per the reviewer's suggestion, we delete the sentence and the text is updated to list specific objectives of the work. Also see response above. CM: p4, I1-8

*p3, l28: The results should be presented, then there should be a separate Discussion section.* AC: As reviewer suggested, we added a separate Discussion section. CM: From p16, l1 to p18, l34

*p4, l27: there has also been some work on phase discrimination using multiple radar wavelengths (e.g., Al-Sakka et al., 2013).*

AC: Thank you for pointing to a relevant work. We use and cite this paper in the revised manuscript.

CM: p18, I17 and I22

*p4 I29 to p5 I3: This paragraph can be reduced to 1-2 sentences, as this information is generally known.*

AC: We update the paragraph to make it more succinct. CM: p3, I15-19

p5, I12-13: please reconsider "nonparametric approaches ... weakness is that the method is sensitive to outliers." I am not sure that this is correct. Parametric approaches tend to be sensitive to outliers.

AC: Nonparametric approaches result in "local" biases as a result of outliers but the effect on global attributes is limited. On the other hand, parametric alternatives can be impacted globally due to biases in the estimated parameters. We reword the sentence in the revised manuscript. CM: p3, I28-29

*p5, I18-20: these two foreshadowing equations are not necessary.* AC / CM: The two foreshadowing sentences have been removed in the revised manuscript.

p7, I4: "classification of precipitation phase at gauge level" is good, but don't we need the phase of precipitation at the radar height to select the appropriate radar Z-R equation? Although this is what Koistinen et al. (2004) and Saltikoff et al. (2015), it doesn't necessarily make it correct. AC: We agree with the reviewer that phase of precipitation at radar measurement height can be different from estimated phase at gauge height. We responded above in detail.

Figure 1: a) I assume that the "length of the observations" is the number of hours with precipitation? b) I also assume that the hypsometry curve is cumulative % of stations below the specified elevation. Please be specific. c) the font size is small and difficult to read. Enlarge and also don't use grey. d) are the red names local cities? Are they important? If so, move them so they are legible.

AC: a) Yes, it is the number of hours with precipitation b) Yes, elevation of gauge locations, c) Font size is increased to make them readable, d) Yes, the cities' names are not important, and the figure and text are updated.

CM: p4, Fig. 1

p7, I12-13: What is the "accumulated hourly radar precipitation rate product?" Is this accumulated from sub-hourly to yield an hourly total, or is the hourly data added? AC: This is accumulated from sub-hourly to yield an hourly total. CM: p5, I24-25 p7, 114: tell us how many gauges in the "a relatively dense network of precipitation gauges." AC: We have changed the text in the revised manuscript. CM: p5, I27

p8, I26: instead of "are" use past tense through the methods. AC / CM: It is corrected.

p8, I34: reword the last sentence "However, we used data from all available precipitation gauges for this study." Perhaps state something about the total number of gauge hours of data used (likely in the order of 100,000 gauge-hours). AC: Nearly 103000 total gauge hours were used in this study. Text is updated with total gauge hours.

CM: p7, 114

p9, I2: provide a source for the "gridded hourly temperature and wind speed dataset" AC: The source for gridded hourly temperature is observations from the Norwegian meteorological stations (refer p9, 17-8.)

The source for gridded wind speed data is the "NORA10" (documented in Reistad et al. (2011)) and "AROME" 2.5 km (documented in Müller et al. (2017)).

Text is updated with source for gridded wind speed data. See response to comment 6) above for more detail.

CM: p6, Table 1

p9, 16: delete "more details on the procedure adopted for catch correction are provided in the next sub-section." AC/ CM: This sentence is deleted.

p9, I10: change "resulted" to "resulting" AC: "resulted" is replaced with "resulting". CM: p6, I11-12

p9, I29: how little is "intensities below 0.1 mmh-1 contributes little?"

AC: The percentage (quantity) is not mentioned in the cited work (Engeland et al., 2014). However, analysis of the data used in this study showed that intensities lower than 0.1 mmh-1 and greater than 0.05 mmh-1 are nearly 10 % of the total data above 0.05 mmh-1. CM: p7, I8-10

p10, I2-8: this is background. It could be moved to earlier in the text, as this is the methods/data section. Tell us what was done. This sentence could also be deleted. AC / CM: As reviewer suggested, we have deleted the sentence.

p10, 14: the word "Nordic" is not necessary here, as it could also be relevant in southern environments

AC / CM: We agree. "Nordic" has been deleted in the revised manuscript.

p10, I4-5: the end of the sentence is redundant "due to large catch errors for snow." In could state that it is "due to high wind conditions." It is wind that causes undercatch, not "large catch errors" AC: Text is reworded.

CM: p7, 115-16

p10, I7: "Wolff et al., 2015" is not in the citation list AC: References is updated with "Wolff et al. (2015)" CM: p22, I35

p10, I9 or previous: what type of precipitation gauge and what type of shield are used? This is very important information to assess the degree of undercatch and the error associated with the undercatch correction.

AC: "The gauges in the study site consists of "Geonor" type and tipping bucket. Both types are with an Alter wind shield. This information is added to the manuscript. CM: p5, I27-28

*p10, I9-17: throughout this paragraph it is "undercatch" correction. This should be consistent, as there are other errors.*

AC: We agree. Text is updated. CM: p7, I20-28

*p10, I13-14: "It was found that correlation between the corrected precipitation by using measured wind speed data (15-gauge locations) and gridded data are over 0.97..." Does this mean the correlation undercatch correction using gauge wind speed versus the downscaled gridded wind speed?*

AC: Yes, the correlation between undercatch corrected precipitation using gauge wind speed versus undercatch corrected precipitation using the downscaled gridded wind speed. This was done to verify that gridded wind would provide a realistic correction compared to wind measured at the site.

p10, 114: there are only 15-gauge locations with wind speed measurements. How are the other 53 precipitation gauges corrected for undercatch? From my comment above (p10, 113-14), I assume that the downscaled gridded wind speed was used to correct for undercatch at all 68 precipitation gauges. This is not clear.

AC: The downscaled gridded wind speed was used to correct for undercatch at all 68 precipitation gauges. To control the result of correcting with gridded wind speed, we compared the corrected precipitation using gridded wind speed with the 15 locations where we had wind speed measurements at the gauge site.

CM: p7, I23-26

p10, I19-22: delete this paragraph. We know what you are going to do Figure 2: add tick marks to the y-axis.

AC: The foreshadowing paragraph is deleted. Tick marks are added to the y-axis. CM: p10, Fig. 2

p10, I23 through p11: Are the "Partial weight of predictors" constant over time, i.e., is there a specific value for station that does not change? AC: No, we did not use a fixed value for any given station. The value varies with the data.

Table 1: As this is a summary of Figure 2, this table could be converted to two horizontals box and whisker plots on Figure 1. The reader doesn't really care about the specific numbers, just the range.

AC: As the reviewer suggested we have plotted box and whisker plot (refer Fig. I in Appendix). However, we feel Table is a better representation of the information we wish to convey and we prefer to keep the Table 1 in the revised manuscript.

p12, I1 and elsewhere: the word "prediction" implies that this is for the future. I suggest using "estimation" throughout. AC: "Prediction is replaced with "estimation".

CM: p10, l4

Figure 3: the caption is confusing; break into two sentences. Also, are these "length of the data (circle size)" the same as in Figure 1? If so, then don't display again here, unless this is relevant later?

AC: The caption is reworded.

Yes, Length of the data (circle size) is same on both Fig. 1 and Fig. 3. Fig.3 is updated. CM: p12, Fig.3

*p13, I4-7: these three sentences could be reduced to a small histogram of RMSE reductions that is added to figure 3.*

AC: As the reviewer suggested, we have plotted a histogram (refer Fig. II in Appendix). However, we prefer to keep these sentences for simplicity in the revised manuscript as well.

p12 to 15: why was a single average (mean) partial weight (beta P = 0.68) used in the k-nn prediction model, when it was shown (Figure 2 and Table 1) that the partial weight varies from 0.4 to 1?

AC: We responded above in detail.

However, the use of station specific partial weight can be more precise when sufficient past observed data are available at each gauge location.

*p15* section 5.3: all this text except for the last sentence is background or methods and should be moved to an appropriate location earlier in the paper.

AC: We think that the first paragraph of section 5.3 present information on the data that is relevant for the understanding of the low intensity analysis and figure 5. We would therefore rather keep this section as it is in the paper.

*p15, I15: does this "still significant" has a statistical meaning? If so, explain how. If not don't use the word significant.*

AC: Yes, the results are statistically significant as the RMSE was estimated using leave one out cross validation (LOOCV). We clarify this in the revised manuscript. CM: p13, I6-8

p15, section 5.4: what is the range of the "Temperature Classes?" You only discuss T > 10C. What about T

**Figure I.** Box and whisker plot of estimated partial weight of predictor variables (Radar precipitation rate and air temperature) at 68 gauge locations in the study area. The summation of partial weights is equal to 1.